# Reducing Adversarially Robust Learning to Non-Robust PAC Learning

**Omar Montasser**
omar@ttic.edu

**Steve Hanneke**
steve.hanneke@gmail.com

**Nathan Srebro**
nati@ttic.edu

Toyota Technological Institute at Chicago

## Abstract

We study the problem of reducing adversarially robust learning to standard PAC learning, i.e. the complexity of learning adversarially robust predictors using access to only a black-box non-robust learner. We give a reduction that can robustly learn any hypothesis class $\mathcal{C}$ using any non-robust learner $\mathcal{A}$ for $\mathcal{C}$. The number of calls to $\mathcal{A}$ depends logarithmically on the number of allowed adversarial perturbations per example, and we give a lower bound showing this is unavoidable.

## 1  Introduction

We consider the problem of learning predictors that are *robust* to adversarial examples at test time. That is, we would like to be robust against an adversary $\mathcal{U} : \mathcal{X} \to 2^{\mathcal{X}}$ that can perturb examples at test-time, where $\mathcal{U}(x) \subseteq \mathcal{X}$ is the set of allowed corruptions the adversary might replace $x$ with, as measured by the *robust risk*:

$$\mathrm{R}_{\mathcal{U}}(\hat{h}; \mathcal{D}) \triangleq \mathop{\mathbb{E}}_{(x,y)\sim\mathcal{D}} \left[ \sup_{z\in\mathcal{U}(x)} \mathbb{1}[\hat{h}(z) \neq y] \right]. \tag{1}$$

For example, $\mathcal{U}$ could be perturbations of bounded $\ell_p$-norms [Goodfellow et al., 2015].

We ask whether we can adversarialy robustly learn a given target hypothesis class $\mathcal{C} \subseteq \mathcal{Y}^{\mathcal{X}}$ (e.g. neural networks)—that is, whether, if there exists a predictor in $\mathcal{C}$ with zero robust risk w.r.t. some unknown distribution $\mathcal{D}$ over $\mathcal{X} \times \mathcal{Y}$, can we find a predictor with (arbitrarily small) robust risk using $m$ i.i.d. (uncorrupted) samples $S = \{(x_i, y_i)\}_{i=1}^{m}$ from $\mathcal{D}$. Recently, Montasser et al. [2019] showed that if $\mathcal{C}$ is PAC-learnable non-robustly, then $\mathcal{C}$ is also adversarially robustly learnable. However, their result is not constructive and the robust learning algorithm given is inefficient, complex, and does not actually directly use a non-robust learner. In this paper, we ask a more constructive version of this question:

*Can we learn adversarially robust predictors given only black-box access to a non-robust learner?*

That is, we are asking whether it is possible to reduce adversarially robust learning to standard non-robust learning. Since we have a plethora of algorithms devised for standard non-robust learning, it would be useful if we could design efficient *reduction* algorithms that leverage such non-robust learning algorithms in a black-box manner to learn *robustly*. That is, design generic wrapper methods that take as input a learning algorithm $\mathcal{A}$ and a specification of the adversary $\mathcal{U}$, and robustly learn by calling $\mathcal{A}$. Many systems in practice perform standard learning but with no robustness guarantees, and therefore, it would be beneficial to provide wrapper procedures that can guarantee adversarial robustness in a black-box manner without needing to modify current learning systems internally.

**Related Work** Recent work [Mansour et al., 2015, Feige et al., 2015, 2018, Attias et al., 2019] can be interpreted as giving reduction algorithms for adversarially robust learning. Specifically, Feige et al. [2015] gave a reduction algorithm that can robustly learn a *finite* hypothesis class $\mathcal{C}$ using black-box access to an ERM for $\mathcal{C}$. Later, Attias et al. [2019] improved this to handle *infinite* hypothesis classes $\mathcal{C}$. But their complexity and the number of calls to ERM depend super-linearly on the number of possible perturbations $|\mathcal{U}| = \sup_x |\mathcal{U}(x)|$, which is undesirable for most types of perturbations—we completely avoid a sample complexity dependence on $|\mathcal{U}|$, and reduce the oracle complexity to at most a poly-logarithmic dependence. Furthermore, their work assumes access specifically to an ERM procedure, which is a very specific type of learner, while we only require access to any method that PAC-learns $\mathcal{C}$ and whose image has bounded VC-dimension.

A related goal was explored by Salman et al. [2020]: They proposed a method to *robustify pre-trained predictors*. Their method takes as input a black-box *predictor* (not a learning algorithm) and a point $x$, and outputs a label prediction $y$ for $x$ and a radius $r$ such that the label $y$ is robust to $\ell_2$ perturbations of radius $r$. But this doesn't guarantee that the predictions $y$ are correct, nor that the radius $r$ would be what we desire, and even if the predictor was returned by a learning algorithm and has a very small non-robust error, we do not end up with any gurantee on the robust risk of the robustified predictor. In this paper, we require black-box access to a *learning algorithm* (not just to a single predictor), but we output a predictor that *is* guaranteed to have *small* robust risk (if one exists in the class, see Definition 2.2). We also provide a general treatment for arbitrary adversaries $\mathcal{U}$, not just $\ell_p$ perturbations.

Finally, we note that the approach of Montasser et al. [2019] can be interpreted as using black-box access to an oracle $\mathrm{RERM}_{\mathcal{C}}$ minimizing the robust *empirical* risk:

$$\hat{h} \in \mathrm{RERM}_{\mathcal{C}}(S) \triangleq \operatorname*{argmin}_{h \in \mathcal{C}} \frac{1}{m} \sum_{i=1}^{m} \sup_{z \in \mathcal{U}(x)} \mathbb{1}[h(z) \neq y]. \tag{2}$$

But this goes well beyond just a *non-robust* learning algorithm, or even ERM.

**Efficient Reductions** From a computational perspective, the relationship between standard non-robust learning and adversarially robust learning is not well-understood. It is natural to wonder whether there is a general efficient reduction for adversarially robust learning, using only non-robust learners. Recent work has provided strong evidence that this is not the case in general. Specifically, Bubeck et al. [2019] showed that there exists a learning problem that can be learned efficiently non-robustly, but is computationally intractable to learn robustly (under plausible complexity-theoretic assumptions). In this paper, we aim to understand when such efficient reductions *are* possible.

**Main Results** When studying reductions of adversarially robust learning to non-robust learning, an important aspect emerges regarding the form of access that the reduction algorithm has to the adversary $\mathcal{U}$. How should we model access to the sets of adversarial perturbations represented by $\mathcal{U}$?

In Section 3, we study the setting where the reduction algorithm has explicit access/knowledge of the possible adversarial perturbations induced by the adversary $\mathcal{U}$ on the *training examples*. We first show that there is an algorithm that can learn adversarially robust predictors with black-box oracle access to a non-robust algorithm:

**Theorem 3.1** (Informal). *For any adversary $\mathcal{U}$, Algorithm 1 robustly learns any target class $\mathcal{C}$ using any black-box non-robust PAC learner $\mathcal{A}$ for $\mathcal{C}$, with $O(\log^2 |\mathcal{U}|)$ oracle calls to $\mathcal{A}$ and sample complexity independent of $|\mathcal{U}|$.*

The oracle complexity dependence on $|\mathcal{U}|$, even if only logarithmic, might be disappointing, but we show it is unavoidable:

**Theorem 3.2** (Informal). *There exists an adversary $\mathcal{U}$ such that for any reduction algorithm $\mathcal{B}$, there exists a target class $\mathcal{C}$ and a PAC learner $\mathcal{A}$ for $\mathcal{C}$ such that $\Omega(\log |\mathcal{U}|)$ oracle queries to $\mathcal{A}$ are necessary to robustly learn $\mathcal{C}$.*

This tells us that only requiring a non-robust PAC learner $\mathcal{A}$ is not enough to avoid the $\log |\mathcal{U}|$ dependence, even with explicit knowledge of $\mathcal{U}$. In Section 4, we show that having an *online* learner $\mathcal{A}$ for $\mathcal{C}$, allows us to robustly learn $\mathcal{C}$ with access to a mistake oracle for $\mathcal{U}$ (see Definition 4.1) where no explicit knowledge of $\mathcal{U}$ is assumed and no dependence on $|\mathcal{U}|$ is incurred:

**Theorem 4.2** (Informal). *There exists an algorithm $\mathcal{B}$ that can robustly learn any target class $\mathcal{C}$ w.r.t. any adversary $\mathcal{U}$ when given access to a mistake oracle $O_{\mathcal{U}}$ and a black-box online learner $\mathcal{A}$ for $\mathcal{C}$. The sample complexity, number of calls to $\mathcal{A}$, and number of calls to $O_{\mathcal{U}}$ are independent of $|\mathcal{U}|$.*

## 2 Preliminaries

Let $\mathcal{X}$ denote the instance space and $\mathcal{Y} = \{\pm 1\}$ denote the label space. Let $\mathcal{U} : \mathcal{X} \to 2^{\mathcal{X}}$ denote an arbitrary adversary. For any adversary $\mathcal{U}$, denote by $|\mathcal{U}| \triangleq \sup_x |\mathcal{U}(x)|$ the number of allowed adversarial perturbations. We start with formalizing the notions of non-robust (standard) PAC learning and robust PAC learning:

**Definition 2.1** (PAC Learnability). *A target hypothesis class $\mathcal{C} \subseteq \mathcal{Y}^{\mathcal{X}}$ is said to be PAC learnable if there exists a learning algorithm $\mathcal{A} : (\mathcal{X} \times \mathcal{Y})^* \to \mathcal{Y}^{\mathcal{X}}$ with sample complexity $m(\varepsilon, \delta) : (0,1)^2 \to \mathbb{N}$ such that: for any $\varepsilon, \delta \in (0,1)$, for any distribution $\mathcal{D}$ over $\mathcal{X} \times \mathcal{Y}$, and any target concept $c \in \mathcal{C}$ with zero risk, $\mathrm{err}_{\mathcal{D}}(c) = 0$, with probability at least $1 - \delta$ over $S \sim \mathcal{D}^{m(\varepsilon, \delta)}$,*

$$\mathrm{err}_{\mathcal{D}}(\mathcal{A}(S)) \triangleq \Pr_{(x,y) \sim \mathcal{D}} [\mathcal{A}(S)(x) \neq y] \leq \varepsilon.$$

**Definition 2.2** (Robust PAC Learnability). *A target hypothesis class $\mathcal{C} \subseteq \mathcal{Y}^{\mathcal{X}}$ is said to be robustly PAC learnable with respect to adversary $\mathcal{U}$ if there exists a learning algorithm $\mathcal{B} : (\mathcal{X} \times \mathcal{Y})^* \to \mathcal{Y}^{\mathcal{X}}$ with sample complexity $m(\varepsilon, \delta) : (0,1)^2 \to \mathbb{N}$ such that: for any $\varepsilon, \delta \in (0,1)$, for any distribution $\mathcal{D}$ over $\mathcal{X} \times \mathcal{Y}$, and any target concept $c \in \mathcal{C}$ with zero robust risk, $\mathrm{R}_{\mathcal{U}}(c; \mathcal{D}) = 0$, with probability at least $1 - \delta$ over $S \sim \mathcal{D}^{m(\varepsilon, \delta)}$,*

$$\mathrm{R}_{\mathcal{U}}(\mathcal{B}(S); \mathcal{D}) \leq \epsilon.$$

We recall the Vapnik-Chervonenkis dimension (VC dimension) is defined as follows:

**Definition 2.3** (VC dimension). *We say that a sequence $\{x_1, \ldots, x_k\} \in \mathcal{X}$ is shattered by $\mathcal{C}$ if $\forall y_1, \ldots, y_k \in \mathcal{Y}, \exists h \in \mathcal{C}$ such that $\forall i \in [k], h(x_i) = y_i$. The VC dimension of $\mathcal{C}$ (denoted $\mathrm{vc}(\mathcal{C})$) is then defined as the largest integer $k$ for which there exists $\{x_1, \ldots, x_k\} \in \mathcal{X}$ that is shattered by $\mathcal{C}$. If no such $k$ exists, then $\mathrm{vc}(\mathcal{C})$ is said to be infinite.*

Another important complexity measure that is utilized in the study of robust PAC learning is the notion of *dual* VC dimension, which we define below:

**Definition 2.4** (Dual VC dimension). *Consider a dual space $\mathcal{G}$: a set of functions $g_x : \mathcal{C} \to \mathcal{Y}$ defined as $g_x(h) = h(x)$, for each $h \in \mathcal{C}$ and each $x \in \mathcal{X}$. Then, the dual VC dimesion of $\mathcal{C}$ (denoted $\mathrm{vc}^*(\mathcal{C})$) is defined as the VC dimension of $\mathcal{G}$. In other words, $\mathrm{vc}^*(\mathcal{C}) = \mathrm{vc}(\mathcal{G})$ and it represents the largest set $\{h_1, \ldots, h_k\}$ that is shattered by points in $\mathcal{X}$.*

If the VC dimension is finite, then so is the dual VC dimesion, and it can be bounded as $\mathrm{vc}^*(\mathcal{C}) < 2^{\mathrm{vc}(\mathcal{C})+1}$ [Assouad, 1983]. Although this exponential dependence is tight for some classes, for many natural classes, such as linear predictors and some neural networks (see, e.g. Lemma 3.2), the primal and dual VC dimensions are equal, or at least polynomially related.

We also formally define what we mean by a *reduction* algorithm:

**Definition 2.5** (Reduction Algorithm). *For an adversary $\mathcal{U}$, a reduction algorithm $\mathcal{B}_{\mathcal{U}}$ takes as input a black-box learning algorithm $\mathcal{A}$ and a training set $S \subseteq \mathcal{X} \times \mathcal{Y}$, and can use $\mathcal{A}$ by calling it $T$ times on inputs $\mathcal{B}_{\mathcal{U}}$ constructs each of size $m_0 \in \mathbb{N}$, and outputs a predictor $f \in \mathcal{Y}^{\mathcal{X}}$.*

We emphasize that $\mathcal{B}_{\mathcal{U}}$ is allowed to be adaptive in its calls to $\mathcal{A}$. That is, it can call $\mathcal{A}$ on one constructed data set, then construct another data set depending on the returned predictor, and call $\mathcal{A}$ on this new data set. Such adaptive use of the base learner $\mathcal{A}$ is central to boosting-type constructions.

We know that a hypothesis class $\mathcal{C}$ is PAC learnable if and only if its VC dimension is finite [Vapnik and Chervonenkis, 1971, 1974, Blumer et al., 1989, Ehrenfeucht et al., 1989]. And in this case, $\mathcal{C}$ is properly PAC learnable with $\mathrm{ERM}_{\mathcal{C}}$. Montasser et al. [2019, Theorem 4] showed that if $\mathcal{C}$ is PAC learnable, then $\mathcal{C}$ is adversarially robustly PAC learnable with an improper learning rule that required a $\mathrm{RERM}_{\mathcal{C}}$ oracle (see Equation 2) and sample complexity of $\tilde{O}\left(\frac{\mathrm{vc}(\mathcal{C})\mathrm{vc}^*(\mathcal{C})}{\varepsilon}\right)$. In this paper, we study whether it is possible to adversarially robustly PAC learn $\mathcal{C}$ using only a black-nox non-robust PAC learner $\mathcal{A}$ for $\mathcal{C}$. We will not require $\mathcal{A}$ is "proper"' (i.e. returns a predictor in $\mathcal{C}$), but we will rely on it returning a predictor in some, possibly much larger, class which still has finite VC-dimension. To this end, we denote by $\mathrm{vc}(\mathcal{A}) = \mathrm{vc}(\mathrm{im}(\mathcal{A}))$ and $\mathrm{vc}^*(\mathcal{A}) = \mathrm{vc}^*(\mathrm{im}(\mathcal{A}))$ the primal and dual VC dimension of the image of $\mathcal{A}$, i.e. the class $\mathrm{im}(\mathcal{A}) = \{\mathcal{A}(S) | S \in (\mathcal{X} \times \mathcal{Y})^*\}$ of the possible hypothesis $\mathcal{A}$ might return. For ERM, or any other proper learner, $\mathrm{im}(\mathcal{A}) \subseteq \mathcal{C}$ and so $\mathrm{vc}(\mathcal{A}) \leq \mathrm{vc}(\mathcal{C})$ and $\mathrm{vc}^*(\mathcal{A}) \leq \mathrm{vc}^*(\mathcal{C})$.

# 3 Learning with Explicitly Specified Adversarial Perturbations

When studying reductions of adversarially robust PAC learning to non-robust PAC learning, an important aspect emerges regarding the form of access that the reduction algorithm has to the adversary $\mathcal{U}$. How should we model access to the sets of adversarial perturbations represented by $\mathcal{U}$?

In this section, we explore the setting where the reduction algorithm has explicit knowledge of the adversary $\mathcal{U}$. That is, the reduction algorithm knows the set of possible adversarial perturbations for each example in the training set. This is in accordance with what is typically considered in practice, where the adversary $\mathcal{U}$ (e.g. $\ell_\infty$ perturbations) is known to the algorithm, and this knowledge is used in adversarial training (see e.g. Madry et al. [2018]). Formally, we consider the following question:

> For any adversary $\mathcal{U}$, does there exist an algorithm that can learn a target class $\mathcal{C}$ *robustly* w.r.t $\mathcal{U}$ given only a black-box non-robust PAC learner $\mathcal{A}$ for $\mathcal{C}$?

We give a positive answer to this question. In Theorem 3.1, we present an algorithm (see Algorithm 1)—based on the $\alpha$-Boost algorithm [Schapire and Freund, 2012, Section 6.4.2] and recent work of Montasser et al. [2019, Theorem 4]— that can adversarially robustly PAC learn a target class $\mathcal{C}$ with only black-box oracle access to a PAC learner $\mathcal{A}$ for $\mathcal{C}$.

---

**Algorithm 1:** Robustify The Non-Robust

**Input:** Training dataset $S = \{(x_1, y_1), \ldots, (x_m, y_m)\}$, black-box non-robust learner $\mathcal{A}$

1   Inflate dataset $S$ to $S_{\mathcal{U}} = \bigcup_{i \leq m} \{(z, y_i) : z \in \mathcal{U}(x_i)\}$.    `// S_U contains all possible perturbations of S.`
2   Set $m_0 = O(\mathrm{vc}(\mathcal{A})\mathrm{vc}(\mathcal{A})^* \log \mathrm{vc}(\mathcal{A})^*)$, and $T = O(\log |S_{\mathcal{U}}|)$.
3   **for** $1 \leq t \leq T$ **do**
4      Set distribution $D_t$ on $S_{\mathcal{U}}$ as in the $\alpha$-Boost algorithm.
5      Sample $S' \sim D_t^{m_0}$, and project $S'$ to dataset $L \subseteq S$ by replacing each perturbation $z$ with its corresponding example $x$.
6      Call `ZeroRobustLoss` on $L$, and denote by $f_t$ its output predictor.
7   Sample $N_{\mathrm{co}} = O\left(\mathrm{vc}^*(\mathcal{A}) \log \mathrm{vc}^*(\mathcal{A})\right)$ i.i.d. indices $i_1, \ldots, i_{N_{\mathrm{co}}} \sim \mathrm{Uniform}(\{1, \ldots, T\})$.
8     (repeat previous step until $f = \mathrm{MAJ}(f_{i_1}, \ldots, f_{i_{N_{\mathrm{co}}}})$ has $R_{\mathcal{U}}(f; S) = 0$)
   **Output:** A majority-vote $\mathrm{MAJ}(f_{i_1}, \ldots, f_{i_{N_{\mathrm{co}}}})$ predictor.

9   `ZeroRobustLoss`(*Dataset L, Learner $\mathcal{A}$*):
10      Inflate dataset $L$ to $L_{\mathcal{U}} = \bigcup_{(x,y) \in L} \{(z, y) : z \in \mathcal{U}(x)\}$, and set $T_L = O(\log |L_{\mathcal{U}}|)$
11      Run $\alpha$-Boost with black-box access to $\mathcal{A}$ on $L_{\mathcal{U}}$ for $T_L$ rounds.
12      Let $h_1, \ldots, h_{T_L}$ denote the hypotheses produced by $\alpha$-Boost with $T_L$ oracle queries to $\mathcal{A}$.
13      Sample $N = O\left(\mathrm{vc}^*(\mathcal{A})\right)$ i.i.d. indices $i_1, \ldots, i_N \sim \mathrm{Uniform}(\{1, \ldots, T_L\})$.
14      (repeat previous step until $f = \mathrm{MAJ}(h_{i_1}, \ldots, h_{i_N})$ has $R_{\mathcal{U}}(f; L) = 0$)
15      **return** $f = \mathrm{MAJ}(h_{i_1}, \ldots, h_{i_N})$

16   $\alpha$-`Boost`(*Dataset L, Learner $\mathcal{A}$*):
17      Initialize $D_1$ to be uniform over $L$, and set $T_L = O(\log |L|)$.
18      **for** $1 \leq t \leq T_L$ **do**
19        Run $\mathcal{A}$ on $S' \sim D_t^{m_0}$, and denote by $h_t$ its output. (repeat until $\mathrm{err}_{D_t}(h_t) \leq 1/3$)
20        Compute a new distribution $D_{t+1}$ by applying the following update for each $(x, y) \in L$:

$$D_{t+1}(x) = \frac{D_t(x)}{Z_t} \times \begin{cases} e^{-2\alpha} & \text{if } h_t(x) = y \\ 1 & \text{otherwise} \end{cases}$$

         where $Z_t$ is a normalization factor and $\alpha$ is set as in Lemma 3.3
21      **return** $h_1, \ldots, h_{T_L}$.

---

**Theorem 3.1.** *For any adversary $\mathcal{U}$, Algorithm 1 can* robustly PAC learn *any target class $\mathcal{C}$ using black-box oracle calls to any PAC learner $\mathcal{A}$ for $\mathcal{C}$ with:*

     *1. Sample Complexity* $m = O\left(\frac{dd^{*2} \log^2 d^*}{\varepsilon} \log\left(\frac{dd^{*2} \log^2 d^*}{\varepsilon}\right) + \frac{\log(1/\delta)}{\varepsilon}\right)$,

     *2. Oracle Complexity* $T = O\left(\left(\log m + \log |\mathcal{U}|\right)^2 + \log(1/\delta)\right)$,

*where $d = \text{vc}(\mathcal{A})$ and $d^* = \text{vc}^*(\mathcal{A})$ are the primal and dual VC dimension of $\mathcal{A}$.*

Importantly, the sample complexity of Algorithm 1 is independent of the number of allowed perturbations $|\mathcal{U}|$, in contrast to work by Attias et al. [2019], that can be interpreted as giving a reduction with sample complexity $m \propto |\mathcal{U}| \log |\mathcal{U}|$, and oracle complexity $T \propto |\mathcal{U}| \log^2 |\mathcal{U}|$.

Before proceeding with the proof of Theorem 3.1, we briefly describe our strategy and its main ingredients. Given a dataset $S$ that is robustly realizable by some target concept $c \in \mathcal{C}$, we show that we can use the non-robust learner $\mathcal{A}$ to implement a RERM oracle that guarantees zero *empirical* robust loss on $S$ using `ZeroRobustLoss` in Algorithm 1. But what about the *population* robust loss? Our main goal to is adversarially robustly learn $\mathcal{C}$ and not just minimize the empirical robust loss. Fortunately, we show that the arguments on *robust* generalization based on sample compression in [Montasser et al., 2019, Theorem 4] will still go through when we replace the $\text{RERM}_{\mathcal{C}}$ oracle they used with our `ZeroRobustLoss` procedure in Algorithm 1. This is achieved by showing that the image of `ZeroRobustLoss` has bounded VC dimension and *dual* VC dimension. The following lemma, whose proof is provided in Appendix A , bounds the *dual* VC dimension of the convex-hull of a class $\mathcal{H}$. This result might be of independent interest.

**Lemma 3.2.** *Let $\text{co}^k(\mathcal{H}) = \{x \mapsto \text{MAJ}(h_1, \ldots, h_k)(x) : h_1, \ldots, h_k \in \mathcal{H}\}$. Then, the dual VC dimension of $\text{co}^k(\mathcal{H})$ satisfies $\text{vc}^*(\text{co}^k(\mathcal{H})) \leq O(d^* \log k)$.*

In addition, we state two extra key lemmas that will be useful for us in the proof. First, Lemma 3.3 states that running $\alpha$-Boost on a dataset for enough rounds produces a sequence of predictors that achieve zero loss on the dataset (with a margin).

**Lemma 3.3** (see, e.g., Corollary 6.4 and Section 6.4.3 in Schapire and Freund [2012]). *Let $S = \{(x_i, c(x_i))\}_{i=1}^m$ be a dataset where $c \in \mathcal{C}$ is some target concept, and $\mathcal{A}$ an arbitrary PAC learner for $\mathcal{C}$ (for $\varepsilon = 1/3$, $\delta = 1/3$). Then, running $\alpha$-Boost (see description in Algorithm 1) on $S$ with black-box oracle access to $\mathcal{A}$ with $\alpha = \frac{1}{2} \ln \left(1 + \sqrt{\frac{2 \ln m}{T}}\right)$ for $T = \lceil 112 \ln(m) \rceil = O(\log m)$ rounds suffices to produce a sequence of hypotheses $h_1, \ldots, h_T \in \text{im}(\mathcal{A})$ such that*

$$\forall (x, y) \in S, \frac{1}{T} \sum_{i=1}^T \mathbb{1}[h_i(x) = y] \geq \frac{5}{9}.$$

*In particular, this implies that the majority-vote $\text{MAJ}(h_1, \ldots, h_T)$ achieves zero error on $S$.*

Second, Lemma 3.4 describes a sparsification technique due to Moran and Yehudayoff [2016] which allows us to control the complexity of the majority-vote predictors that we use in Algorithm 1.

**Lemma 3.4** (Sparsification of Majority Votes, Moran and Yehudayoff [2016]). *Let $\mathcal{H}$ be a hypothesis class with finite primal and dual VC dimension, and $h_1, \ldots, h_T$ be predictors in $\mathcal{H}$. Then, for any $(\varepsilon, \delta) \in (0, 1)$, with probability at least $1 - \delta$ over $N = O\left(\frac{\text{vc}^*(\mathcal{H}) + \log(1/\delta)}{\varepsilon^2}\right)$ independent random indices $i_1, \ldots, i_N \sim \text{Uniform}(\{1, \ldots, T\})$, we have:*

$$\forall (x, y) \in \mathcal{X} \times \mathcal{Y}, \left| \frac{1}{N} \sum_{j=1}^N \mathbb{1}[h_{i_j}(x) = y] - \frac{1}{T} \sum_{i=1}^T \mathbb{1}[h_i(x) = y] \right| < \varepsilon.$$

We are now ready to proceed with the proof of Theorem 3.1.

*Proof of Theorem 3.1.* Let $\mathcal{U}$ be an arbitrary adversary. Let $\mathcal{C}$ be a target class that is PAC learnable with some PAC learner $\mathcal{A}$. Let $\mathcal{H}$ denote the base class of hypotheses of learner $\mathcal{A}$. Let $d$ denote the VC dimension of $\mathcal{H}$, and $d^*$ denote the dual VC dimension of $\mathcal{H}$. Our proof is divided into two parts.

**Zero Empirical Robust Loss.** Let $L = \{(x_1, y_1), \ldots, (x_m, y_m)\}$ be a dataset that is *robustly* realizable with some target concept $c \in \mathcal{C}$; in other words, for each $(x, y) \in L$ and each $z \in \mathcal{U}(x)$, $c(z) = y$. We will show that we can use the non-robust learner $\mathcal{A}$ to guarantee zero *empirical* robust loss on $L$. This procedure is described in `ZeroRobustLoss` in Algorithm 1. We inflate dataset $L$ to include all possible perturbations under the adversary $\mathcal{U}$. Let $L_{\mathcal{U}} = \bigcup_{i \leq m} \{(z, y_i) : z \in \mathcal{U}(x_i)\}$ denote the inflated dataset. Observe that $|L_{\mathcal{U}}| \leq m|\mathcal{U}|$, since each point $x \in \mathcal{X}$ has at most $|\mathcal{U}|$

possible perturbations. We run the $\alpha$-Boost algorithm on the inflated dataset $L_{\mathcal{U}}$ with *black-box* access to PAC learner $\mathcal{A}$, where in each round of boosting $m_0$ samples are fed to $\mathcal{A}$ (where $m_0$ is chosen Step 2). By Lemma 3.3, running $\alpha$-Boost with $T = O\left(\log(|L_{\mathcal{U}}|)\right)$ oracle calls to $\mathcal{A}$ suffices to produce a sequence of hypotheses $h_1, \ldots, h_T \in \mathcal{H}$ such that

$$\forall (z, y) \in L_{\mathcal{U}}, \frac{1}{T} \sum_{i=1}^{T} \mathbb{1}[h_i(z) = y] \geq \frac{5}{9}.$$

Specifically, the above implies that a majority-vote over hypotheses $h_1, \ldots, h_T$ achieves zero *robust loss* on dataset $L$, $\mathrm{R}_{\mathcal{U}}(\mathrm{MAJ}(h_1, \ldots, h_T); L) = 0$. By Step 13 in `ZeroRobustLoss` in Algorithm 1 and Lemma 3.4 (with $\varepsilon = 1/18, \delta = 1/3$), we have that for $N = O(d^*)$, the sampled predictors $h_{i_1}, \ldots, h_{i_N}$ satisfy

$$\forall (z, y) \in L_{\mathcal{U}}, \frac{1}{N} \sum_{j=1}^{N} \mathbb{1}[h_{i_j}(z) = y] > \frac{1}{T} \sum_{i=1}^{T} \mathbb{1}[h_i(z) = y] - \frac{1}{18} > \frac{5}{9} - \frac{1}{18} = \frac{1}{2}.$$

Therefore, the majority-vote over the sampled hypotheses $\mathrm{MAJ}(h_{i_1}, \ldots, h_{i_N})$ achieves zero robust loss on $L$, $\mathrm{R}_{\mathcal{U}}(\mathrm{MAJ}(h_{i_1}, \ldots, h_{i_N}); S) = 0$. Thus, we can implement a RERM oracle (see Equation 2) using the procedure `ZeroRobustLoss` in Algorithm 1. The sparsification step (Step 12) controls the complexity of the image of `ZeroRobustLoss`, i.e., the hypothesis class that is being implicitly used. Specifically, observe that the sparsified predictor $f = \mathrm{MAJ}(h_{i_1}, \ldots, h_{i_N})$ lives in $\mathrm{co}^{O(d^*)}(\mathcal{H})$, which is the convex-hull of $\mathcal{H}$ that combines at most $O(d^*)$ predictors. To guarantee *robust* generalization in the next part, it suffices to bound the VC dimension and dual VC dimension of $\mathrm{co}^{O(d^*)}(\mathcal{H})$. By [Blumer et al., 1989], the VC dimension of $\mathrm{co}^{O(d^*)}(\mathcal{H})$ is at most $O(dd^* \log d^*)$, and by Lemma 3.2, the dual VC dimension of $\mathrm{co}^{O(d^*)}(\mathcal{H})$ is at most $O(d^* \log d^*)$.

**Robust Generalization through Sample Compression.** This part builds on the approach of Montasser et al. [2019, Theorem 4]. Specifically, we observe that their proof works even if we replace the $\mathrm{RERM}_{\mathcal{C}}$ oracle they used, with our `ZeroRobustLoss` procedure in Algorithm 1 that is described above. We provide a self-contained analysis below.

Let $\mathcal{D}$ be an arbitrary distribution over $\mathcal{X} \times \mathcal{Y}$ that is robustly realizable with some concept $c \in \mathcal{C}$, i.e., $\mathrm{R}_{\mathcal{U}}(c; \mathcal{D}) = 0$. Fix $\varepsilon, \delta \in (0, 1)$ and a sample size $m$ that will be determined later. Let $S = \{(x_1, y_1), \ldots, (x_m, y_m)\}$ be an i.i.d. sample from $\mathcal{D}$. We run the $\alpha$-Boost algorithm (see Algorithm 1) on the inflated dataset $S_{\mathcal{U}}$, this time with `ZeroRobustLoss` as the subprocedure. Specifically, on each round of boosting, $\alpha$-Boost computes an empirical distribution $D_t$ over $S_{\mathcal{U}}$ (according to Step 18). We draw $m_0 = O(dd^* \log d^*)$ samples $S'$ from $D_t$, and *project* $S'$ to a dataset $L_t \subset S$ by replacing each perturbation $(z, y) \in S'$ with its corresponding original point $(x, y) \in S$, and then we run `ZeroRobustLoss` on dataset $L_t$. The projection step is crucial for the proof to work, since we use a *sample compression* argument to argue about *robust* generalization, and the sample compression must be done on the *original* points that appeared in $S$ rather than the perturbations in $S_{\mathcal{U}}$.

By classic PAC learning guarantees [Vapnik and Chervonenkis, 1974, Blumer et al., 1989], with $m_0 = O(\mathrm{vc}(\mathrm{co}^{O(d^*)}(\mathcal{H}))) = O(dd^* \log d^*)$, we are guaranteed uniform convergence of 0-1 risk over predictors in $\mathrm{co}^{O(d^*)}(\mathcal{H})$ (the effective hypothesis class used by `ZeroRobustLoss`). So, for any distribution $D$ over $\mathcal{X} \times \mathcal{Y}$ with $\inf_{c \in \mathcal{C}} \mathrm{err}(c; \mathcal{D}) = 0$, with nonzero probability over $S' \sim \mathcal{D}^{m_0}$, every $f \in \mathrm{co}^{O(d^*)}(\mathcal{H})$ satisfying $\mathrm{err}_{S'}(f) = 0$, also has $\mathrm{err}_D(f) < 1/3$. By the guarantee of `ZeroRobustLoss` (established above), we know that $f_t = \texttt{ZeroRobustLoss}(L_t, \mathcal{A})$ achieves zero robust loss on $L_t$, $\mathrm{R}_{\mathcal{U}}(f_t; L_t) = 0$, which by definition of the projection means that $\mathrm{err}_{S'}(f_t) = 0$, and thus $\mathrm{err}_{D_t}(f_t) < 1/3$. This allows us to use `ZeroRobustLoss` with $\alpha$-Boost to establish a *robust* generalization guarantee. Specifically, Lemma 3.3 implies that running the $\alpha$-Boost algorithm with $S_{\mathcal{U}}$ as its dataset for $T = O(\log(|S_{\mathcal{U}}|))$ rounds, using `ZeroRobustLoss` to produce the hypotheses $f_t \in \mathrm{co}^{O(d^*)}(\mathcal{H})$ for the distributions $D_t$ produced on each round of the algorithm, will produce a sequence of hypotheses $f_1, \ldots, f_T \in \mathrm{co}^{O(d^*)}(\mathcal{H})$ such that:

$$\forall (z, y) \in S_{\mathcal{U}}, \frac{1}{T} \sum_{i=1}^{T} \mathbb{1}[f_i(z) = y] \geq \frac{5}{9}.$$

Specifically, this implies that the majority-vote over hypotheses $f_1, \ldots, f_T$ achieves zero *robust* loss on dataset $S$, $\mathrm{R}_{\mathcal{U}}(\mathrm{MAJ}(f_1, \ldots, f_T); S) = 0$. Note that each of these classifiers $f_t$ is equal to `ZeroRobustLoss`$(L_t, \mathcal{A})$ for some $L_t \subseteq S$ with $|L_t| = m_0$. Thus, the classifier $\mathrm{MAJ}(f_1, \ldots, f_T)$ is representable as the value of an (order-dependent) reconstruction function $\phi$ with a compression set size

$$m_0 T = O(\mathrm{vc}(\mathrm{co}^{O(d^*)}(\mathcal{H})) \log(|S_{\mathcal{U}}|)) = O(dd^* \log d^* \, (\log m + \log |\mathcal{U}|)).$$

This is not enough, however, to obtain a sample complexity bound that is independent of $|\mathcal{U}|$. For that, we will sparsify the majority-vote as in Step 7 in Algorithm 1. Lemma 3.4 (with $\varepsilon = 1/18, \delta = 1/3$) guarantees that for $N_{\mathrm{co}} = O(d^* \log d^*)$, the sampled predictors $f_{i_1}, \ldots, f_{i_{N_{\mathrm{co}}}}$ satisfy:

$$\forall (z, y) \in S_{\mathcal{U}}, \frac{1}{N_{\mathrm{co}}} \sum_{j=1}^{N_{\mathrm{co}}} \mathbb{1}[f_{i_j}(z) = y] > \frac{1}{T} \sum_{i=1}^{T} \mathbb{1}[f_i(z) = y] - \frac{1}{18} > \frac{5}{9} - \frac{1}{18} = \frac{1}{2},$$

so that the majority-vote achieves zero robust loss on $S$, $\mathrm{R}_{\mathcal{U}}(\mathrm{MAJ}(f_{i_1}, \ldots, f_{i_{N_{\mathrm{co}}}}); S) = 0$. Since again, each $f_{i_j}$ is the result of `ZeroRobustLoss`$(L_t, \mathcal{A})$ for some $L_t \subseteq S$ with $|L_t| = m_0$, we have that the classifier $\mathrm{MAJ}(f_{i_1}, \ldots, f_{i_{N_{\mathrm{co}}}})$ can be represented as the value of an (order-dependent) reconstruction function $\phi$ with a compression set size $m_0 N_{\mathrm{co}} = O(dd^* \log d^* \cdot d^* \log d^*) = O(dd^{*2} \log^2(d^*))$. Lemma A.1 (Montasser et al. [2019]) which extends to the robust loss the classic compression-based generalization guarantees from the 0-1 loss, implies that for $m \geq cdd^{*2} \log^2(d^*)$ (for an appropriately large numerical constant $c$), with probability at least $1 - \delta$ over $S \sim \mathcal{D}^m$,

$$R_{\mathcal{U}}(\mathrm{MAJ}(f_{i_1}, \ldots, f_{i_{N_{\mathrm{co}}}}); \mathcal{D}) \leq O\left( \frac{dd^{*2} \log^2(d^*)}{m} \log(m) + \frac{1}{m} \log(1/\delta) \right).$$

Setting this less than $\varepsilon$ and solving for a sufficient size of $m$ to achieve this yields the stated sample complexity bound.

Our oracle complexity $T$ (number of calls to $\mathcal{A}$) is at most $O((\log |S_{\mathcal{U}}|)^2 + \log(1/\delta)) \leq O\left( (\log m + \log |\mathcal{U}|)^2 + \log(1/\delta) \right)$, since `ZeroRobustLoss` in Algorithm 1 terminates in at most $O(\log |S_{\mathcal{U}}|)$ rounds each time it is invoked, and it is invoked at most $O(\log |S_{\mathcal{U}}|)$ times by the outer $\alpha$-Boost algorithm in Algorithm 1. Therefore, we have at most $O\left( (\log m + \log |\mathcal{U}|)^2 \right)$ geometric random variables that represent that number of times Step 19 is invoked, which is the step where learner $\mathcal{A}$ is called. The success probability of Step 19 is a constant (say $2/3$), therefore the mean of the sum of the geometric random variables is $O\left( (\log m + \log |\mathcal{U}|)^2 \right)$. Since sums of geometric random variables concentrate around the mean [Brown], we get that with probability at least $1 - \delta$, the total number of times Step 19 is executed is at most $O\left( (\log m + \log |\mathcal{U}|)^2 + \log(1/\delta) \right)$. This concludes the proof. $\qquad\square$

## 3.1 A Lowerbound on the Oracle Complexity

The oracle complexity of Algorithm 1 depends on $\log |\mathcal{U}|$. Can this dependence be reduced or avoided? Unfortunately, we show in Theorem 3.5 that the dependence on $\log |\mathcal{U}|$ is unavoidable and *no* reduction algorithm can do better.

It is relatively easy to show, by relying on lower bounds from the boosting literature [Schapire and Freund, 2012, Section 13.2.2], that for any reduction algorithm, there exists a target class $\mathcal{C}$ with $\mathrm{vc}(\mathcal{C}) \leq 1$ and a PAC learner $\mathcal{A}$ for $\mathcal{C}$ such that $\Omega(\log |\mathcal{U}|)$ oracle calls to this PAC learner are necessary to achieve zero *empirical* robust loss. But this is done by constructing a "crazy" improper learner with $\mathrm{vc}(\mathcal{A}) \propto |\mathcal{U}| \gg \mathrm{vc}(\mathcal{C})$.

Perhaps we can ensure better oracle complexity by requiring a more constrained PAC learner $\mathcal{A}$, e.g. with low VC dimension (as in our upper bound), or perhaps even a proper learner, or an ERM, or maybe where $\mathrm{im}(\mathcal{A})$ (and so also $\mathcal{C}$) is finite. We next present a lower bound to show that none of these help improve the oracle complexity. Specifically, we will present a construction showing that for any reduction algorithm $\mathcal{B}$ there is a randomized target class $\mathcal{C}$ and a PAC learner $\mathcal{A}$ for $\mathcal{C}$ with

$\mathrm{vc}(\mathcal{A}) = 1$ where $\mathcal{B}$ needs to make $\Omega(\log |\mathcal{U}|)$ oracle calls to $\mathcal{A}$ to robustly learn $\mathcal{C}$. The idea here is that the target class $\mathcal{C}$ is chosen randomly after $\mathcal{B}$, and so $\mathcal{B}$ essentially knows nothing about $\mathcal{C}$ and needs to communicate with $\mathcal{A}$ in order to learn. As a reminder, a reduction algorithm has a budget of $T$ oracle calls to a non-robust learner $\mathcal{A}$, where each oracle call is constructed with $m_0$ points, or more generally a *distribution* over $\mathcal{X} \times \mathcal{Y}$. We next show that any successful reduction requires $T = \Omega(\log |\mathcal{U}|)$ for some non-robust learner $\mathcal{A}$ (proof provided in Appendix A).

**Theorem 3.5.** *For any sufficiently large integer $u$, if $|\mathcal{X}| \geq u^{10u}$, there exists an adversary $\mathcal{U}$ with $|\mathcal{U}| = u$ such that for any reduction algorithm $\mathcal{B}$ and for any $\varepsilon > 0$, there exists a target class $\mathcal{C}$ and a PAC learner $\mathcal{A}$ for $\mathcal{C}$ with $\mathrm{vc}(\mathcal{A}) = 1$ such that, if the training sample has size at most $(1/8)|\mathcal{U}|^9$, then $\mathcal{B}$ needs to make $T \geq \frac{\log|\mathcal{U}|}{\log(2/\varepsilon)}$ oracle calls to $\mathcal{A}$ in order to robustly learn $\mathcal{C}$.*

**Computational Efficiency** Although the sample complexity of Algorithm 1 is independent of $|\mathcal{U}|$, we showed that the $\log |\mathcal{U}|$ dependence in oracle complexity is unavoidable. This implies that the runtime of Algorithm 1 will be at best weakly polynomial and have at least a $\log |\mathcal{U}|$ dependence. But maybe this is not so bad, because it is equivalent to the number of bits required to represent the adversarial perturbations. This weak poly-time dependence is common in almost all optimization algorithms (gradient descent, interior point methods, etc). What is more concerning is the linear runtime and memory dependence on $|\mathcal{U}|$ that emerges from the explicit representation of the adversarial perturbations during training. In practice, many of the adversarial perturbations $\mathcal{U}$ are infinite, but specified implicitly, and not by enumerating over all possible perturbations (e.g. $\ell_p$ perturbations). This motivates the following next steps: What operations do we need to be able to implement efficiently on $\mathcal{U}$ in order to robustly learn? What access (oracle calls, or "interface") do we need to $\mathcal{U}$?

**Sampling Oracle Over Perturbations** A reasonable form of access to $\mathcal{U}$ that is sufficient for implementing Algorithm 1 is a sampling oracle that takes as input a point $x$ and an energy function $E : \mathcal{X} \to \mathbb{R}$, and does the following:

(a) Samples a perturbation $z$ from a distribution given by $p_x(z) \propto \exp(E(z)) * \mathbb{1}[z \in \mathcal{U}(x)]$. That is, the oracle samples from the set $\mathcal{U}(x)$ based on the weighting encoded in $E$.
(b) Calculates $\Pr[z \in \mathcal{U}(x)]$ for the distribution given by $p(z) \propto \exp(E(z))$.

With such an oracle, Algorithm 1 can be implemented without the need to do explicit inflation of $S$ to $S_{\mathcal{U}}$, and can avoid the linear dependence on $|\mathcal{U}|$. This is because Algorithm 1 and its subprocedure `ZeroRobustLoss` just need to sample from distributions over the inflated set $S_{\mathcal{U}}$ that are constructed by $\alpha$-Boost (as required in Steps 5 and 17 in Algorithm 1). This can be simulated via a two-stage process where we maintain a conditional distribution over $S$ (the original points), and then draw perturbations using the sampling oracle. Specifically, to sample from a distribution $D_t$ that is constructed by $\alpha$-Boost, we use two energy functions $E_t^+(z) = -2\alpha \sum_{i \leq t} \mathbb{1}[g_t(z) = +1]$ and $E_t^-(z) = -2\alpha \sum_{i \leq t} \mathbb{1}[g_t(z) = -1]$, where $g_1, \ldots, g_t$ represent the sequence of predictors produced during the first $t$ rounds of boosting (either $h_t$'s produced by non-robust learner $\mathcal{A}$ or $f_t$'s produced by `ZeroRobustLoss`). Using the sampling oracle, we can sample from $D_t$, by first sampling $(x, y)$ from $S$ based on the marginal estimates computed by the oracle (operation (b) described above) using energy function $E_t^y$, and then sampling $z$'s from their $\mathcal{U}(x)$ (operation (a) described above) using energy function $E_t^y$.

## 4    Learning with a Mistake Oracle for Adversarial Perturbations

In Section 3, we considered an explicit form of access to the set of adversarial perturbations $\mathcal{U}$, as well as access via a sampling oracle. A more realistic form of access is having a mistake oracle for $\mathcal{U}$:

**Definition 4.1** (Mistake Oracle). *Denote by $\mathsf{O}_{\mathcal{U}}$ a mistake oracle for $\mathcal{U}$. $\mathsf{O}_{\mathcal{U}}$ takes as input a predictor $f : \mathcal{X} \to \mathcal{Y}$ and an example $(x, y)$ and either: (a) asserts that $f$ is robust on $(x, y)$ (i.e. $\forall z \in \mathcal{U}(x), f(z) = y$), or (b) returns an adversarial perturbation $z \in \mathcal{U}(x)$ such that $f(z) \neq y$.*

Having only a mistake oracle for $\mathcal{U}$, rather than an explicit representation, is a more realistic form of access. In this case, the reduction algorithm has no explicit knowledge of the set of adversarial perturbations $\mathcal{U}$ and is forced to interact with the mistake oracle $\mathsf{O}_{\mathcal{U}}$ in order to learn an adversarially robust predictor. Furthermore, what is typically referred to as adversarial training in practice fits exactly into this framework [Madry et al., 2018].

Does access to a mistake oracle $O_\mathcal{U}$ suffice to robustly learn a target class $\mathcal{C}$ using a black-box non-robust learner $\mathcal{A}$ for $\mathcal{C}$? First, can we achieve a similar upper bound as in Theorem 3.1 but with a mistake oracle $O_\mathcal{U}$ rather than explicit access to $\mathcal{U}$? Unfortunately, even with a black-box ERM for $\mathcal{C}$, one can show that $|\mathcal{U}|$ oracle calls to $O_\mathcal{U}$ are unavoidable (proof provided in Appendix A):

**Claim 4.2.** *For any reduction algorithm $\mathcal{B}$, there exists an adversary $\mathcal{U}$, target class $\mathcal{C}$, and an* ERM *for $\mathcal{C}$ with VC dimension 1, such that $\mathcal{B}$ needs to make $T \geq |\mathcal{U}|$ oracle calls to $O_\mathcal{U}$.*

Thus, a non-robust PAC learner $\mathcal{A}$ for $\mathcal{C}$ is not enough to learn $\mathcal{C}$ robustly with a mistake oracle $O_\mathcal{U}$ *without a linear dependence on $|\mathcal{U}|$.* This suggests that a stronger assumption about $\mathcal{A}$ is required. We next show that an *online* learner $\mathcal{A}$ for $\mathcal{C}$ suffices to robustly learn $\mathcal{C}$ with a mistake oracle $O_\mathcal{U}$ *and* without any dependence on $|\mathcal{U}|$. Before proceeding, we include a brief reminder of what it means to learn in an online setting with a finite mistake bound:

**Definition 4.3.** *(Mistake Bound Model) We say an online learner $\mathcal{A}$ learns a hypothesis class $\mathcal{C}$ with mistake bound $M_\mathcal{A}$ if learner $\mathcal{A}$ makes at most $M_\mathcal{A}$ mistakes on any sequence of examples that are labeled with some concept $c \in \mathcal{C}$.*

We are now ready to state our main result for this section.

**Theorem 4.4.** *Algorithm 3 robustly PAC learns any target class $\mathcal{C}$ w.r.t. an adversary $\mathcal{U}$ with black-box access to a mistake oracle $O_\mathcal{U}$ and an online learner $\mathcal{A}$ for $\mathcal{C}$ with sample complexity, number of calls to $\mathcal{A}$, and number of calls to $O_\mathcal{U}$ that is at most $2\frac{M_\mathcal{A}}{\varepsilon}\log\left(\frac{M_\mathcal{A}}{\delta}\right)$, where $M_\mathcal{A}$ is the mistake-bound of online learner $\mathcal{A}$.*

*Proof sketch.* Run the online learner $\mathcal{A}$ on the sequence of input examples using the mistake oracle $O_\mathcal{U}$ to find mistakes. Details and algorithm are provided in Appendix A. □

**Example 4.5.** *Let $\mathcal{C}$ be the class of OR functions over the boolean hypercube $\{0,1\}^n$. There is an online learner $\mathcal{A}$ that learns $\mathcal{C}$ with a mistake bound $M_\mathcal{A} = n$. Theorem 4.4 implies that we can robustly learn $\mathcal{C}$ using $\mathcal{A}$ with sample complexity, number of calls to $\mathcal{A}$, and number of calls to $O_\mathcal{U}$ that is at most $2\frac{n}{\varepsilon}\log\left(\frac{n}{\delta}\right)$.*

# 5 Discussion

The main contribution of this paper is in formulating the question of reducing adversarially robust learning to standard non-robust learning and providing answers in some settings. We outline a few directions for furture work below.

**Mistake Oracle for $\mathcal{U}$** This is a more challenging setting (but perhaps more realistic) where the reduction algorithm has no knowledge of $\mathcal{U}$ and can only interact with a mistake oracle for $\mathcal{U}$. Theorem 4.4 shows that online learnability is sufficient for robust learning in this model. Beyond this, are there weaker conditions that would enable robust learning under this model? Or is having an online learner essential? What if we consider specific target classes? Montasser et al. [2020] recently gave an algorithm that robustly learns halfspaces in this model. A natural next step is to ask which other classes can be robustly learned in this model, or more ambitiously characterize a necessary and sufficient condition for learning in this model.

**Agnostic Setting** We focused only on robust PAC learning in the realizable setting, where we assume there is a $c \in \mathcal{C}$ with zero robust error. It would be desirable to extend our results also to the agnostic setting, where we want to compete with the best $c \in \mathcal{C}$. We remark that an agnostic-to-realizable reduction described in Montasser et al. [2019, Theorem 6] can be used in our setting, however, it has runtime that is exponential in $vc(\mathcal{A})$. Another attempt through the agnostic boosting frameworks [e.g. Kalai and Kanade, 2009] requires a non-robust PAC learner $\mathcal{A}$ with error $\varepsilon$ that scales with $|\mathcal{U}|^2$, which results in a sample complexity that depends on $|\mathcal{U}|$, and this is something we would like to avoid.

**Boosting and Robustness** Boosting has led to many exciting developments in theory and practice of machine learning. It started with asking: Can we boost the accuracy of weak predictors to attain a predictor with high accuracy? Freund and Schapire [1997] showed that boosting the accuracy is possible and can be done efficiently. What we consider in this paper can be viewed as a question of boosting robustness: Can we boost non-robust predictors to attain a *robust* predictor? and can we do this efficiently? Another natural question to consider which we did not study in this paper is: Can we boost *weakly* robust predictors to attain a *robust* predictor?

## Broader Impact

Learning predictors that are robust to adversarial perturbations is an important challenge in contemporary machine learning. Current machine learning systems have been shown to be brittle against different notions of robustness such as adversarial perturbations [Szegedy et al., 2013, Biggio et al., 2013, Goodfellow et al., 2014], and there is an ongoing effort to devise methods for learning predictors that *are* adversarially robust. As machine learning systems become increasingly integrated into our everyday lives, it becomes crucial to provide guarantees about their performance, even when they are used outside their intended conditions.

We already have many tools developed for standard learning, and having a universal *wrapper* that can take any standard learning method and turn it into a *robust* learning method could greatly simplify the development and deployment of learning that is *robust* to test-time adversarial perturbations. The results that we present in this paper are still mostly theoretical, and limited to the realizable setting, but we expect and hope they will lead to further theoretical study as well as practical methodological development with direct impact on applications.

In this work we do not deal with training-time adversarial attacks, which is a major, though very different, concern in many cases.

As with any technology, having a more robust technology can have positive and negative societal consequences, and this depends mainly on how such technology is utilized. Our intent from this research is to help with the design of robust machine learning systems for application domains such as healthcare and transportation where its critical to ensure performance guarantees even outside intended conditions. In situations where there is a tradeoff between robustness and accuracy, this work might be harmful in that it would prioritize robustness over accuracy and this may not be ideal in some application domains.

## Acknowledgments and Disclosure of Funding

We would like to thank Shay Moran for the insightful discussions that led to the formalization of the question we study in this paper. We also thank the anonymous reviewers for their thoughtful and helpful feedback. This work is partially supported by DARPA[1] cooperative agreement HR00112020003.

## Footnotes

[1]This paper does not reflect the position or the policy of the Government, and no endorsement should be inferred.

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
