[Supplementary Material]

# A Appendix

*Proof of Lemma 3.2.* Consider a *dual space* $\bar{\mathcal{G}}$: a set of functions $\bar{g}_x : \text{co}^k(\mathcal{H}) \to \mathcal{Y}$ defined as $\bar{g}_x(f) = f(x)$ for each $f = \text{MAJ}(h_1, \ldots, h_k) \in \text{co}^k(\mathcal{H})$ and each $x \in \mathcal{X}$. It follows by definition of dual VC dimension that $\text{vc}(\bar{\mathcal{G}}) = \text{vc}^*(\text{co}^k(\mathcal{H}))$. Similarly, define another dual space $\mathcal{G}$: a set of functions $g : \mathcal{H} \to \mathcal{Y}$ defined as $g(x) = h(x)$ for each $h \in \mathcal{H}$ and each $x \in \mathcal{X}$. We know that $\text{vc}(\mathcal{G}) = \text{vc}^*(\mathcal{H}) = d^*$. Observe that by definition of $\mathcal{G}$ and $\bar{\mathcal{G}}$, we have that for each $x \in \mathcal{X}$ and each $f = \text{MAJ}(h_1, \ldots, h_k) \in \text{co}^k(\mathcal{H})$,

$$\bar{g}_x(f) = f(x) = \text{MAJ}(h_1, \ldots, h_k)(x) = \text{sign}\left(\sum_{i=1}^{k} h_i(x)\right) = \text{sign}\left(\sum_{i=1}^{k} g_x(h_i)\right).$$

By the Sauer-Shelah Lemma applied to dual class $\mathcal{G}$, for any set $H = \{h_1, \ldots, h_n\} \subseteq \mathcal{H}$, the number of possible behaviors

$$|\mathcal{G}|_H| := |\{(g_x(h_1), \ldots, g_x(h_n)) : x \in \mathcal{X}\}| \leq \binom{n}{\leq d^*}. \tag{3}$$

Consider a set $F = \{f_1, \ldots, f_m\} \subseteq \text{co}^k(\mathcal{H})$, the number of possible behaviors can be upperbounded as follows:

$$
\begin{aligned}
\left|\bar{\mathcal{G}}|_F\right| &= |\{(\bar{g}_x(f_1), \ldots, \bar{g}_x(f_m)) : x \in \mathcal{X}\}| \\
&= \left|\{(\bar{g}_x(\text{MAJ}(h_1^1, \ldots, h_1^k)), \ldots, \bar{g}_x(\text{MAJ}(h_m^1, \ldots, h_m^k))) : x \in \mathcal{X}\}\right| \\
&= \left|\left\{\left(\text{sign}\left(\sum_{i=1}^{k} g_x(h_i)\right), \ldots, \text{sign}\left(\sum_{i=1}^{k} g_x(h_i)\right)\right) : x \in \mathcal{X}\right\}\right| \\
&\overset{(i)}{\leq} \left|\{(g_x(h_1^1), \ldots, g_x(h_1^k), g_x(h_2^1), \ldots, g_x(h_2^k), \ldots, g_x(h_m^1), \ldots, g_x(h_m^k)) : x \in \mathcal{X}\}\right| \\
&\overset{(ii)}{\leq} \binom{mk}{\leq d^*},
\end{aligned}
$$

where $(i)$ follows from observing that each expanded vector $(g_x(h_i^1), \ldots, g_x(h_i^k))_{i=1}^m \in \mathcal{Y}^{mk}$ can map to at most one vector $\left(\text{sign}\left(\sum_{i=1}^{k} g_x(h_i)\right), \ldots, \text{sign}\left(\sum_{i=1}^{k} g_x(h_i)\right)\right) \in \mathcal{Y}^m$, and $(ii)$ follows from Equation 3. Observe that if $\left|\bar{\mathcal{G}}|_F\right| < 2^m$, then by definition, $F$ is not shattered by $\bar{\mathcal{G}}$, and this implies that $\text{vc}(\bar{\mathcal{G}}) < m$. Thus, to conclude the proof, we need to find the smallest $m$ such that $\binom{mk}{\leq d^*} < 2^m$. It suffices to check that $m = O(d^* \log k)$ satisfies this condition. $\qquad\square$

**Lemma A.1** (Montasser et al. [2019]). *For any $k \in \mathbb{N}$ and fixed function $\phi : (\mathcal{X} \times \mathcal{Y})^k \to \mathcal{Y}^{\mathcal{X}}$, for any distribution $P$ over $\mathcal{X} \times \mathcal{Y}$ and any $m \in \mathbb{N}$, for $S = \{(x_1, y_1), \ldots, (x_m, y_m)\}$ iid $P$-distributed random variables, with probability at least $1 - \delta$, if $\exists i_1, \ldots, i_k \in \{1, \ldots, m\}$ s.t. $\hat{R}_{\mathcal{U}}(\phi((x_{i_1}, y_{i_1}), \ldots, (x_{i_k}, y_{i_k})); S) = 0$, then*

$$R_{\mathcal{U}}(\phi((x_{i_1}, y_{i_1}), \ldots, (x_{i_k}, y_{i_k})); P) \leq \frac{1}{m - k}(k \ln(m) + \ln(1/\delta)).$$

*Proof of Theorem 3.5.* We begin with describing the construction of the adversary $\mathcal{U}$. Let $m \in \mathbb{N}$; we will construct $\mathcal{U}$ with $|\mathcal{U}| = 2^m$, supposing $|\mathcal{X}| \geq 2\binom{2^{10m}}{2^m} + 2^{10m}$. Let $Z = \{z_1, \ldots, z_{2^{10m}}\} \subset \mathcal{X}$ be a set of $2^{10m}$ unique points from $\mathcal{X}$. For each subset $L \subset Z$ where $|L| = 2^m$, pick a unique pair $x_L^+, x_L^- \in \mathcal{X} \setminus Z$ and define $\mathcal{U}(x_L^+) = \mathcal{U}(x_L^-) = L$. That is, for every choice $L$ of $2^m$ perturbations from $Z$, there is a corresponding pair $x_L^+, x_L^-$ where $\mathcal{U}(x_L^+) = \mathcal{U}(x_L^-) = L$. For any point $x \in \mathcal{X} \setminus Z$ that is remaining, define $\mathcal{U}(x) = \{\}$.

Let $\mathcal{B}$ be an arbitrary reduction algorithm, and let $\varepsilon > 0$ be the error requirement. We will now describe the construction of the target class $\mathcal{C}$. The target class $\mathcal{C}$ will be constructed randomly. Namely, we will first define a labeling $\tilde{h} : Z \to \mathcal{Y}$ on the perturbations in $Z$ that is positive on the first half of $Z$ and negative on the second half of $Z$: $\tilde{h}(z_i) = +1$ if $i \leq \frac{2^{10m}}{2}$, and $\tilde{h}(z_i) = -1$ if

$i > \frac{2^{10m}}{2}$. Divide the positive/negative halves into groups of size $2^m$:

$$\underbrace{\{\text{first } 2^m \text{ positives}\}}_{G_1^+}, \ldots, \underbrace{\{\text{last } 2^m \text{ positives}\}}_{G_{2^{9m-1}}^+} \Big| \underbrace{\{\text{first } 2^m \text{ negatives}\}}_{G_1^-}, \ldots, \underbrace{\{\text{last } 2^m \text{ negatives}\}}_{G_{2^{9m-1}}^-}.$$

Let $\varepsilon' = \varepsilon/2$. The target concept $h^* : \mathcal{X} \to \mathcal{Y}$ is generated by randomly flipping the labels of an $\varepsilon'$ fraction of the points in each group $G_1^+, \ldots, G_{2^{9m-1}}^+$ from positive to negative and randomly flipping the labels of an $\varepsilon'$ fraction of the points in each group $G_1^-, \ldots, G_{2^{9m-1}}^-$ from negative to positive. This defines $h^*$ on $Z$; then for every pair $x^+, x^- \in \mathcal{X} \setminus Z$ where $\mathcal{U}(x^+) = \mathcal{U}(x^-) \neq \{\}$, define $h^*(x^+) = +1$ and $h^*(x^-) = -1$. Once $h^*$ is generated, we define the distribution $D_{h^*}$ over $\mathcal{X} \times \mathcal{Y}$ that will be used in the lower bound by swapping the $\varepsilon'$ fractions of points with the flipped labels in each pair $(G_1^+, G_1^-), \ldots, (G_{2^{9m-1}}^+, G_{2^{9m-1}}^-)$ which defines new positive/negative pairs: $(G(h^*)_1^+, G(h^*)_1^-), \ldots, (G(h^*)_{2^{9m-1}}^+, G(h^*)_{2^{9m-1}}^-)$. Let $x_i^+, \_ = \mathcal{U}^{-1}(G(h^*)_i^+)$ and $\_, x_i^- = \mathcal{U}^{-1}(G(h^*)_i^-)$ for each $i \in [2^{9m-1}]$ ($\mathcal{U}^{-1}$ returns a pair of points). Observe that by definition of $h^*$ on $\mathcal{X} \setminus Z$, we have that $h^*(x_i^+) = +1$ and $h^*(x_i^-) = -1$ since $h^*(z) = +1 \forall z \in G(h^*)_i^+$ and $h^*(z) = -1 \forall z \in G(h^*)_i^-$. Let $D_{h^*}$ be a uniform distribution over $(x_1^+, +1), (x_1^-, -1), \ldots, (x_{2^{9m-1}}^+, +1), (x_{2^{9m-1}}^-, -1)$.

Let $T \leq \frac{\log 2^m}{\log(1/\varepsilon')}$. Define a randomly-constructed target class $\mathcal{C} = \{h_1, \ldots, h_T, h_{T+1}\}$ where $h_{T+1} = h^*$ and $h_1, h_2, \ldots, h_T$ are generated according the following process: If $t = 1$, then $h_1 := \tilde{h}$ (augmented to all of $\mathcal{X}$ by letting $\tilde{h}(x) = h^*(x)$ for all $x \in \mathcal{X} \setminus Z$). For $t \geq 2$, let $\text{DIS}_{t-1} = \{z \in Z : h_{t-1}(z) \neq h^*(z)\}$, and construct $h_t$ by flipping a uniform randomly-selected $1 - \varepsilon'$ fraction of the labels of $h_{t-1}$ in $G_i^+ \cap \text{DIS}_{t-1}$ and $1 - \varepsilon'$ fraction of the labels of $h_{t-1}$ in $G_i^- \cap \text{DIS}_{t-1}$ for each $i \in [2^{9m-1}]$. Observe that by construction, $h_1, \ldots, h_T$ satisfy the property that they agree with $h^*$ on $\mathcal{X} \setminus Z$, i.e. $h_t(x) = h^*(x)$ for each $t \leq T$ and each $x \in \mathcal{X} \setminus Z$.

We now state a few properties of the randomly-constructed target class $\mathcal{C}$ that we will use in the remainder of the proof. First, observe that by definition of $\text{DIS}_t$ for $t \leq T$, we have that $G_i^{\pm} \cap \text{DIS}_T \subseteq G_i^{\pm} \cap \text{DIS}_{T-1} \subseteq \cdots \subseteq G_i^{\pm} \cap \text{DIS}_1$ for each $1 \leq i \leq 2^{9m-1}$. In addition,

$$|G_i^{\pm} \cap \text{DIS}_t| \geq \varepsilon' |G_i^{\pm} \cap \text{DIS}_{t-1}| \text{ for each } 1 \leq i \leq 2^{9m-1}.$$

By the random process generating $h^*$, we also know that $|G_i^{\pm} \cap \text{DIS}_1| \geq \varepsilon' 2^m$. Combined with the above, this implies that:

$$|G_i^{\pm} \cap \text{DIS}_T| \geq \varepsilon'^T 2^m \text{ for each } 1 \leq i \leq 2^{9m-1}.$$

So, for $T \leq \frac{\log 2^m}{\log(2/\varepsilon)}$, we are guaranteed that $|G_i^{\pm} \cap \text{DIS}_T| \geq 1$ for each $1 \leq i \leq 2^{9m-1}$.

We now describe the construction of a PAC learner $\mathcal{A}$ with $\text{vc}(\mathcal{A}) = 1$ for the randomly generated concept $h^*$ above; we assume that $\mathcal{A}$ knows $\mathcal{C}$ (but of course, $\mathcal{B}$ does not know $\mathcal{C}$).

---

**Algorithm 2:** Non-Robust PAC Learner $\mathcal{A}$

**Input:** Distribution $P$ over $\mathcal{X}$.
**Output:** $h_s$ for the *smallest* $s \in [T]$ with $\text{err}_P(h_s, h^*) \leq \varepsilon$ (or outputting $h_{T+1} = h^*$ if no such $s$ exists).

---

First, we will show that $\text{vc}(\mathcal{A}) = 1$. By definition of $\mathcal{A}$, it suffices to show that $\text{vc}(\mathcal{C}) = \text{vc}(\{h^*, h_1, \ldots, h_T\}) = 1$. By definition of $h^*$ and $h_1$, it is easy to see that there is a $z \in Z$ where $h^*(z) \neq h_1(z)$, and thus $\text{vc}(\mathcal{C}) \geq 1$. Observe that by construction, each predictor in $h_1, \ldots, h_T$ operates as a threshold in each group $G_1^+, G_1^-, \ldots, G_{2^{9m-1}}^+, G_{2^{9m-1}}^-$ (ordered according to the order in which the labels are flipped in the $h_1, \ldots, h_T$ sequence). As a result, each $x \in \mathcal{X}$ has its label flipped at most once in the sequence $(h_1(x), \ldots, h_T(x), h^*(x))$. This is because once the ground-truth label of $x$, $h^*(x)$, is revealed by some $h_t$ (i.e., $h_t(x) = h^*(x)$), all subsequent predictors $h_{t'}$ satisfy $h_{t'}(x) = h^*(x)$. Thus, for any two points $z, z' \in \mathcal{X}$, the number of possible behaviors $|\{(h(z), h(z')) : h \in \mathcal{C}\}| \leq 3$. Therefore, $\mathcal{C}$ cannot shatter two points. This proves that $\text{vc}(\mathcal{C}) \leq 1$.

**Analysis** Suppose that we run the reduction algorithm $\mathcal{B}$ with non-robust learner $\mathcal{A}$ for $T$ rounds to obtain predictors $h_{s_1} = \mathcal{A}(P_1), \ldots, h_{s_T} = \mathcal{A}(P_T)$. We will show that $\Pr_{h^*}[s_T \leq T | S] > 0$,

meaning that with non-zero probability learner $\mathcal{A}$ will not reveal the ground-truth hypothesis $h^*$. For $t \leq T$, let $E_t$ denote the event that $\mathrm{err}_{P_t}(h_{s_{t-1}+1}, h^*) \leq \varepsilon$. When conditioning on $S, s_1, \ldots, s_{t-1}$, observe that by construction of the randomized hypothesis class $\mathcal{C}$, for each $i \leq 2^{9m-1}$ such that $\{(x_i^-, -1), (x_i^+, +1)\} \cap S = \emptyset$, and each $z \in G_i^{\pm} \cap \mathrm{DIS}_{s_{t-1}}$ : $\Pr_{h^*}\left[h^*(z) \neq h_{s_{t-1}+1}(z)|S, s_1, \ldots, s_{t-1}\right] \leq \varepsilon' = \varepsilon/2$. It follows then by the law of total probability that for any distribution $P_t$ constructed by $\mathcal{A}$:

$$\mathbb{E}_{h^*}\left[\mathrm{err}_{P_t}(h_{s_{t-1}+1}, h^*)|S, s_1, \ldots, s_{t-1}\right] \leq \frac{\varepsilon}{2}.$$

By Markov's inequality, it follows that

$$\Pr_{h^*}\left[\bar{E}_t|S, s_1, \ldots, s_{t-1}\right] = \Pr_{h^*}\left[\mathrm{err}_{P_t}(h_{s_{t-1}+1}, h^*) > \varepsilon|S, s_1, \ldots, s_{t-1}\right]$$

$$\leq \frac{\mathbb{E}_{h^*}\left[\mathrm{err}_{P_t}(h_{s_{t-1}+1}, h^*)|S, s_1, \ldots, s_{t-1}\right]}{\varepsilon} \leq \frac{1}{2}.$$

By law of total probability,

$$\Pr_{h^*}\left[s_T \leq T|S\right] \geq \Pr_{h^*}\left[E_1|S\right] \times \Pr_{h^*}\left[E_2|S, E_1\right] \times \cdots \times \Pr_{h^*}\left[E_T|S, E_1, \ldots, E_{T-1}\right] \geq \left(\frac{1}{2}\right)^T > 0.$$

To conclude the proof, we will show that if the reduction algorithm $\mathcal{B}$ sees at most $1/2$ of the support of distribution $D_{h^*}$ through a training set $S$ and makes only $T \leq \frac{\log 2^m}{\log(2/\varepsilon)}$ oracle calls to $\mathcal{A}$, then it will likely fail in robustly learning $h^*$. For each $i \leq 2^{9m-1}$, conditioned on the event that $\{(x_i^-, -1), (x_i^+, +1)\} \cap S = \emptyset$, and conditioned on $h_{s_1}, \ldots, h_{s_T}$, there is a $z \in Z$ that is equally likely to be in $\mathcal{U}(x_i^-)$ or $\mathcal{U}(x_i^+)$. To see why such a point exists, we first describe an equivalent distribution generating $h^*, h_1, \ldots, h_T$. For each $i \leq 2^{9m-1}$ randomly select a $2\varepsilon'$ fraction of points from $G_i^+$ and a $2\varepsilon'$ fraction of points from $G_i^-$. Then, randomly pair the points in each $2\varepsilon'$ fraction to get $\varepsilon' 2^m$ pairs $z_i, z_i'$ for each $G_i^{\pm}$. For each pair $z_i, z_i'$ flip a fair coin $c_i$: if $c_i = 1$, $z_i$'s label gets flipped and otherwise if $c_i = 0$ then $z_i'$'s label gets flipped. This is equivalent to generating $h^*$ by flipping the labels of a uniform randomly-selected $\varepsilon$ fraction of points in each $G_i^{\pm}$ as originally described, but is helpful book-keeping that simplifies our analysis. In addition, $h_1, \ldots, h_T$ can be generated in a similar fashion. Since $T \leq \frac{\log 2^m}{\log(2/\varepsilon)}$, we are guaranteed that $|G_i^{\pm} \cap \mathrm{DIS}_{s_T}| \geq 1$. By definition of $\mathrm{DIS}_{s_T}$, this implies that that there is a pair of points $z_i, z_i'$ in each $G_i^{\pm}$ where each $h_{s_t}(z_i) = h_{s_t}(z_i')$ for $t \leq T$ but $h^*(z_i) \neq h^*(z_i')$ (i.e., each $h_{s_t}$ never reveals the ground-truth label for at least one pair). And then in the end, if $\{(x_i^-, -1), (x_i^+, +1)\} \cap S = \emptyset$, $\mathcal{B}$ will make some prediction on $z_i$, and the posterior probability of it being wrong is $1/2$.

More formally, for any training dataset $S \sim D_{h^*}^{|S|}$ where $|S| \leq 2^{9m-3}$, any $h_{s_1}, \ldots, h_{s_T}$ returned by $\mathcal{A}$ where $T \leq \frac{\log 2^m}{\log(2/\varepsilon)}$, and any predictor $f : \mathcal{X} \to \mathcal{Y}$ that is picked by $\mathcal{B}$:

$$\mathbb{E}_{h^*}\left[R_{\mathcal{U}}(f; D_{h^*})|S, h_{s_1}, \ldots, h_{s_T}\right] \geq \mathbb{E}_{h^*}\left[\frac{1}{2^{9m}} \sum_{\substack{(x,y) \notin S, \\ (x,y) \in \mathrm{supp}(D_{h^*})}} \sup_{z \in \mathcal{U}(x)} \mathbb{1}[f(z) \neq y] \middle| S, h_{s_1}, \ldots, h_{s_T}\right]$$

$$= \frac{1}{2^{9m}} \sum_{i=1}^{2^{9m-1}} \Pr_{h^*}\Big[\big((x_i^+, +1), (x_i^-, -1) \notin S\big) \wedge$$

$$\big(\exists z \in \mathcal{U}(x_i^+) \text{ s.t. } f(z) \neq +1 \vee \exists z \in \mathcal{U}(x_i^-) \text{ s.t. } f(z) \neq -1\big) \middle| S, h_{s_1}, \ldots, h_{s_T}\Big]$$

$$\geq \frac{2^{9m-1}}{2^{9m}}\frac{1}{2} = \frac{1}{4}.$$

This implies that, for any $\mathcal{B}$ limited to $n \leq 2^{9m-3}$ training examples and $T \leq \frac{m}{\log_2(2/\varepsilon)}$ queries, there exists a *deterministic* choice of $h^*$ and $h_1, \ldots, h_T$, and a corresponding learner $\mathcal{A}$ that is a

PAC learner for $\{h^*\}$ using hypothesis class $\{h^*, h_1, \ldots, h_T\}$ of VC dimension 1, such that, for $S \sim D_{h^*}^n$, $\mathbb{E}_S[\mathrm{R}_{\mathcal{U}}(f; D_{h^*})] \geq \frac{1}{4}$. $\qquad\square$

*Proof sketch of Claim 4.2.* Let $\mathcal{B}$ be an arbitrary reduction algorithm. Let $x_0, x_1 \in \mathcal{X}$, and $k \in \mathbb{N}$. Pick arbitrary points $Z = \{z_1, \ldots, z_{2k}\} \subseteq \mathcal{X}$. Let $X = \{x_0, x_1\} \cup Z$. Let $b \in \{0, 1\}^{2k}$ be a bit string drawn uniformly at random from the set $\left\{b \in \{0, 1\}^{2k} : \sum_i b_i = k\right\}$, think of this as a random partition of $Z$ into two equal sets $Z_0$ and $Z_1$. For each $y \in \{0, 1\}$, define $\mathcal{U}_b(x_y)$ to include $x_y$ and all perturbations $z \in Z_y$. Also, foreach $z \in Z$ define $\mathcal{U}_b(z) = \{z\}$. Similarly, define target class $\mathcal{C}_b$ to include only a single hypothesis $c_b$ where $c_b(\mathcal{U}(x_0)) = 0$ and $c_b(\mathcal{U}(x_1)) = 1$. We will consider an ERM that uses the set of thresholds $\mathcal{H}_\phi = \{x \mapsto \mathbb{1}[\phi(x) \geq \theta] : \theta \in \mathbb{R}\}$ as its hypothesis class, where $\phi$ is a random embedding such that for each $z_0 \in \mathcal{U}_b(x_0)$ and each $z_1 \in \mathcal{U}_b(x_1)$: $\phi(z_0) < \phi(z_1)$; this guarantees that the random hypothesis $c_b$ is realized by some $h \in \mathcal{H}_\phi$. On any input $L \subseteq X \times \{0, 1\}$, we define the ERM to return the earliest possible threshold that reveals as few 0's as possible.

Since algorithm $\mathcal{B}$ only sees training data $S = \{(x_0, 0), (x_1, 1)\}$ as its input, by picking $b$ uniformly at random, $\mathcal{B}$ has no way of knowing which perturbations belong to $\mathcal{U}(x_0)$ and which belong to $\mathcal{U}(x_1)$, and therefore its forced to call the mistake oracle $\mathsf{O}_{\mathcal{U}}$ at least $k$ times. The ERM oracle is designed such that it will reveal as little information about this as possible.

Suppose that we run algorithm $\mathcal{B}$ for $T$ rounds, where in each round $t \leq T$, $\mathcal{B}$ maintains a predictor $f_t : X \to \{0, 1\}$ that determines that labeling of $x_0, x_1$ and the set of perturbations $Z$. We will show that, in expectation over the random choice of $b$ and $\phi$, in order for the final predictor $f_T$ outputted by $\mathcal{B}$ to have robust loss zero on $S$, i.e. $\mathrm{R}_{\mathcal{U}_b}(f_T) = 0$, the number of rounds $T$ needs to be at least $k$.

On each round $t \leq T$, $\mathcal{B}$ is allowed to:

1. Query the mistake oracle $\mathsf{O}_{\mathcal{U}}$ with a query consisting of some predictor $g_t : X \to \{0, 1\}$ and a point $(x, y) \in X \times \{0, 1\}$.
2. Query the ERM oracle with a dataset $L_t \subseteq X \times \{0, 1\}$.

Let $M_t = \sum_{z \in Z} \mathbb{1}[f_t(z) \neq c_b(z)]$ be the number of mistakes at round $t$, and let $H_t = \{g_j, (x_j, y_j), L_j\}_{j \leq t}$ denote the history of queries. Then, observe that

$$\mathbb{E}_{b, \phi} [M_t | M_{t-1}, H_{t-1}] \geq M_{t-1} - 1.$$

This is because oracle $\mathsf{O}_{\mathcal{U}}$ reveals the ground truth label of at most 1 point at round $t$, and the ERM will move the threshold by at most one position. This implies that $\mathbb{E}_{b, \phi}[M_T | M_0, H_0] \geq M_0 - T$. We can further condition on the event that $M_0 \geq k$ which has non-zero probability (since $b$ is picked uniformly at random). This implies, by the probabilistic method, that there exists $b, \phi$ such that for $T \leq k - 1$, $M_T \geq 1$. Therefore, by definition of $M_T$, $f_T$ is not be robustly correct on $S$ for $T \leq k - 1$. $\qquad\square$

*Proof of Theorem 4.4.* Let $\mathcal{U}$ be an arbitrary adversary and $\mathsf{O}_{\mathcal{U}}$ its corresponding mistake oracle. Let $\mathcal{C} \subseteq \mathcal{Y}^{\mathcal{X}}$ be an arbitrary target class, and $\mathcal{A}$ an online learner for $\mathcal{C}$ with mistake bound $M_{\mathcal{A}} < \infty$. We assume w.l.o.g. that the online learner $\mathcal{A}$ is conservative, meaning that it does not update its state unless it makes a mistake. Algorithm 3 in essence is a standard conversion of a learner in the mistake bound model to a learner in the PAC model (see e.g. Balcan [2010]):

---

**Algorithm 3:** Robust Learner with a Mistake Oracle.

---

**Input:** $S = \{(x_1, y_1), \ldots, (x_m, y_m)\}$, $\varepsilon, \delta$, black-box access to a an online learner $\mathcal{A}$, black-box access to a mistake oracle $\mathsf{O}_{\mathcal{U}}$

1 Initialize $h_0 = \mathcal{A}(\emptyset)$.
2 **for** $i \leq m$ **do**
3      Certify the robustness of $h$ on $(x_i, y_i)$ by asking the mistake oracle $\mathsf{O}_{\mathcal{U}}$.
4      If $h_t$ is not robust on $(x_i, y_i)$, update $h_t$ by running $\mathcal{A}$ on $(z, y_i)$, where $z$ is the perturbation returned by $\mathsf{O}_{\mathcal{U}}$.
5      Break when $h_t$ is robustly correct on a consecutive sequence of length $\frac{1}{\varepsilon} \log \left(\frac{M_{\mathcal{A}}}{\delta}\right)$.

**Output:** $h_t$.

---

**Analysis** Let $\mathcal{D}$ be an arbitrary distribution over $\mathcal{X} \times \mathcal{Y}$ that is robustly realizable with some concept $c \in \mathcal{C}$,i.e., $R_{\mathcal{U}}(c; \mathcal{D}) = 0$. Fix $\varepsilon, \delta \in (0, 1)$ and a sample size $m = 2\frac{M_{\mathcal{A}}}{\varepsilon} \log\left(\frac{M_{\mathcal{A}}}{\delta}\right)$.

Since online learner $\mathcal{A}$ has a mistake bound of $M_{\mathcal{A}}$, Algorithm 3 will terminate in at most $\frac{M_{\mathcal{A}}}{\varepsilon} \log\left(\frac{M_{\mathcal{A}}}{\delta}\right)$ steps of certification, which of course is an upperbound on the number of calls to the mistake oracle $O_{\mathcal{U}}$, and the number of calls to the online learner $\mathcal{A}$.

It remains to show that the output of Algorithm 3, the final predictor $h$, has low robust risk $R_{\mathcal{U}}(h; \mathcal{D})$. Throughout the runtime of Algorithm 3, the online learner can generate a sequence of at most $M_{\mathcal{A}} + 1$ predictors. There's the initial predictor from Step 1, plus the $M_{\mathcal{A}}$ updated predictors corresponding to potential updates by online learner $\mathcal{A}$. Observe that the probability that the final $h$ has robust risk more than $\varepsilon$

$$\Pr_{S \sim \mathcal{D}^m}[R_{\mathcal{U}}(h; \mathcal{D}) > \varepsilon] \leq \Pr_{S \sim \mathcal{D}^m}[\exists j \in [M_{\mathcal{A}} + 1] \text{ s.t. } R_{\mathcal{U}}(h_j; \mathcal{D}) > \varepsilon] \leq (M_{\mathcal{A}} + 1)(1 - \varepsilon)^{\frac{1}{\varepsilon} \log\left(\frac{M_{\mathcal{A}} + 1}{\delta}\right)} \leq \delta.$$

Therefore, with probability at least $1 - \delta$ over $S \sim \mathcal{D}^m$, Algorithm 3 outputs a predictor $h$ with robust risk $R_{\mathcal{U}}(h; \mathcal{D}) \leq \varepsilon$. Thus, Algorithm 3 robustly PAC learns $\mathcal{C}$ w.r.t. adversary $\mathcal{U}$. $\quad\square$