[Reviews · NeurIPS 2020]

Review 1

Summary and Contributions: The paper provides algorithms for adversarially robust learning via reductions from standard PAC learning. The paper is exclusively theoretical. The authors provide bounds on the sample complexity and oracle complexity (for the adversary) in terms of the primal & dual VC dimensions of the hypothesis class and the size of the perturbation set of the adversary.

Strengths: The overall problem is well motivated. I find the approach of reducing robust learning to standard learning particularly interesting. If successful, it would allow us to leverage a large body of work for non-robust learning.

Weaknesses: The paper presents only theoretical results and it is unclear to what extent the theoretical models capture the empirically interesting phenomena in adversarially robust learning. For instance, the proposed Algorithm 1 relies on inflating the perturbation set, which is not practical for perturbations such as the widely-studied l_inf or l_2 balls. The paper later addresses this problem via an online learning approach. However the efficiency of this approach is still unclear since the paper does not specify the mistake bound of the online learning in Theorem 4.3.

Correctness: The claims seem correct, but it is hard to check correctness because the paper is not self-contained (see the "clarity" points below).

Clarity: Overall the paper is well written, but some parts are unclear. As mentioned above, it would be helpful for the reader to instantiate Theorem 4.3 to get an overall bound. It would also be helpful to make the paper more self-contained, e.g., by providing a full proof of Theorem 3.1 instead of referring to Montasser et al. 2019. Similarly, stating the relevant results from Schapire and Freund, 2012 would make the paper more accessible.

Relation to Prior Work: The relation to prior work is clear.

Reproducibility: Yes

Additional Feedback: Overall I like the direction of this paper. Unfortunately the presentation is not self-contained and the connections to the empirical side of adversarial robustness are unclear, which overall makes me hesitant to recommend acceptance. --------------------------------------------- Comments after the rebuttal: Thank you for addressing my points. Based on the discussion with the other reviewers, I increased my score. While the novelty from an empirical perspective may be limited (e.g., adversarial training has already been combined with perturbations beyond l_p balls in https://arxiv.org/abs/1712.02779 and https://arxiv.org/abs/1902.07906 ), I agree that the theoretical perspective is interesting. It would indeed be helpful to add a concrete instantiation of Theorem 4.3.


Review 2

Summary and Contributions: Robust learning has got a lot of attention over the last decade. This paper is about *test-time attacks (called evasion attacks or attacks finding "adversarial examples"). Such attacks are studied widely from experimental point of view, but more recently theoretical results are being proven for understanding how, and when, learning under such adversarial perturbations are possible. Montasser et al (MHS COLT 19) showed that if a problem is PAC learnable, i.e., has finite VC dimension d, then it is also robustly PAC learnable, though the sample complexity (in their proof) could be as large as exponential(d). This paper's main contribution is an alternative proof of the result of MHS, by showing how to reduce the task of robust learning to the task of normal PAC learning. The reduction is not efficient (i.e., polynomial time) but still is meaningful in a statistical sense. The key ideas are to use a boosting algorithm (known as alpha-boosting) and the compression-based proof of MHS (and some other, perhaps more minor, tools, such as sparsification technique of Moran-Yehudaoff). In fact, alpha-boosting is also used by the MHS work, so at a high level, one can see this work as "re-organizing" the proof of MHS and improve to be a reduction to a general PAC learning rather than a direct proof. In more detail, the new proof consists of two layers of alpha-boosting: the inner layer uses alpha-boosting to implement a robust ERM oracle, using which the outer alpha-boosting procedure is proved effective using the same compression-based arguments of MHS. Other contributions of the paper are as follows: - Paper's proof (i.e., reduction) calls the underlying PAC learning algorithm ~poly(log(U)) where U is the total number of adversarial perturbations allowed to a given point. While U could be seen to be infinite (e.g., in continuous spaces) for practical purposes, one can discretize the perturbations, making them only exponentially many cases, and so log(U) would be polynomial. - The paper also proves a lower bound for this number of calls to be Omega(log(U)). - The paper makes a distinction between how the set of perturbations are affecting the learning process. Currently, the algorithm needs full set of perturbations in its intermediate steps. A natural way to limit this is to have a "mistake oracle" that gets (x,y) and returns a perturbation of x that has a different label than y (hence being a mistake in a robust-inference sense). The paper shows that such oracles are enough to get robust learning, if the problem has a robust *online* learner (with a bounded number of mistakes on any sequence of inputs) to begin with. Post rebuttal: Thanks for clarifications and the plan to address the concerns. The rebuttal indeed had a lot of useful comments that would greatly benefit the paper once included. The most important ones are to make sure that the paper is self contained.

Strengths: Addressing an important basic question in robust learning: I think the main result of the paper is more interesting conceptually, as it does not improve the final result of MHS. That is because PAC learning implies bounded VC dimension and MHS already shows that bounded VC dimension implies robust PAC learnability. Giving a more modular proof for a recent very interesting result: The new proof also makes the result of MHS more understandable and perhaps paves the way to further developments that can be seen more easily in light of the more boosting-based proof. Other contributions (i.e., distinctions on how perturbations are accessed, as well as the reduction for the online setting) are also interesting first steps.

Weaknesses: The technical contribution of the main result (in light of the previous work of Montasser et al) is limited. (Conceptually it stays interesting to get a more modular proof.) The final result is way far from being computationally relevant (some ideas about how to address this aspect is discussed, and also in general this is not the paper's fault that we are far from such computationally efficient results; it is just a hard problem).

Correctness: The paper/proofs are formally written (though too compactly).

Clarity: The presentation is too dense and could improve a lot, e.g., by explaining the alpha-boosting and explicitly saying the properties of it that they need rather than touching upon them during the proof. My score is conditioned on fixing the writing issues.

Relation to Prior Work: Most key related works are mentioned, but quite a few others are yet to be added to a related work section (see the feedback part below).

Reproducibility: Yes

Additional Feedback: The paper misses quite a bit of citations on related work. For example, original papers that formally studied the problem of robust inference/learning also used the same definition, and are missing from related work: - Yishay Mansour, Aviad Rubinstein, and Moshe Tennenholtz. Robust probabilistic inference, 2015 - Uriel Feige, Yishay Mansour, and Robert Schapire. Learning and inference in the presence of corrupted inputs, 2015 - Uriel Feige, Yishay Mansour, and Robert E Schapire. Robust inference for multiclass classification, 2018 Reader seems to be assumed to know the details of alpha-boosting. Please spell out the abstract tools/lemmas/constructions about alpha-boosting that are needed for a modular presentation and proof of your result. The MHS work also discusses the agnostic setting (using known reductions). Can you add more discussions on why same approaches do not work for your setting to cover the non-realizable setting as well? the "Sampling Oracle Over Perturbations" paragraph (line 287) is a very important and interesting sub-topic, but I could not understand the compact paragraph. Can you please elaborate more and write more formally what exact sampling procedures (from the set of perturbations) are enough for the algorithms to go through? Minor comments: "whether its possible" -> whether it's possible (or it is possible) line 89: last word, the exponent has a typo in parentheses line 189, 190: though this sparsification can perhaps be done deterministically (as pointed out in MHS), but the way you do it is by random selection. So, the guarantee of the inequality you state only holds with high probability (rather than with probability 1) When using elps-delta, eg. in line 203 and Theorem 3.4, please state what they are (of course they are PAC learning parameters, but this still should be stated). the notion of "mistake bound" is well-known for online learning, but still for self-readability of statements add a line and define it. In supplemental material, line 5 of Alg 3, it seems you need to use M_A instead of (t+1)^2 (otherwise, I don't understand the proof).


Review 3

Summary and Contributions: SUMMARY This work considers the following question: given oracle access to a non-robust PAC learning algorithm and a specification of all the ways that an adversary could perturb any given point (e.g. the map U taking any image x to the set U(x) of all L_inf-bounded perturbations of x), produce a PAC learner that achieves low *robust* error. They work in the robustly realizable case, i.e. where we're promised that a hypothesis achieving zero population *robust* error exists. Their main results are: 1) Such a reduction exists in the case where |U(x)| \le s for all x. The resulting PAC learner has s-independent sample complexity and makes at most ~log^2 s calls to the non-robust learner. 2) There exist non-robust learners for which log s oracle calls are necessary, i.e. the above problem as stated can be impossible if |U(x)| is infinite for some x. 3) Suppose now we only have *restricted* access to the perturbation sets in the form of a "mistake oracle" that given (x,y) and a predictor either certifies that f is robustly correct on (x,y) or produces a perturbation x' of x for which f(x') \neq y. Then given an online learner with mistake bound M, one can robustly learn with \tilde{O}(M) samples/queries to the learner/queries to the mistake oracle. ------------------------------- CONTEXT Previous work has considered the question of designing "robustifying" wrappers around non-robust learning algorithms, though they assume oracle access to either a particular algorithm, e.g. ERM or robust ERM, and potentially incur worse overhead in sample/oracle complexity, or to a predictor that they try to robustify only in the sense that the prediction should not change under a prescribed set of perturbations, but not in the sense of getting low robust error ------------------------------- TECHNIQUES 1) appears to be their main result, and as the authors note, its proof builds heavily on the proof of Theorem 4 in Montasser-Hanneke-Srebro '19. Both works bound sample complexity for achieving low robust error by exhibiting a sample compression scheme via the following recipe: a) "inflate" the dataset by replacing every (x,y) with all (z,y) for which z\in U(x) (this is slightly more involved in [MHS19] where they considered potentially infinite U(x)'s). b) by robust realizability and standard non-robust generalization bounds, for any distribution D over the inflated dataset, any predictor achieving zero non-robust error on m_0 samples from D has constant advantage wrt D, where m_0 is sufficiently large but s-independent. c) run log(|inflated dataset|) rounds of boosting to get an ensemble of hypotheses whose majority achieves zero error on the inflated dataset and therefore zero empirical robust error on the original dataset d) bring the subset of samples output by the compression scheme down to s-independent size using a sparsification trick of Moran-Yehudayoff The main difference in the present work is that in step b), the predictor used to achieve zero non-robust error on the set of samples of size m_0 was simply a robust ERM oracle. Rather than simply assume access to this powerful subroutine, the main content of the present work is to show how to implement such a subroutine just using access to a non-robust PAC learning algorithm. They do this by cleverly running the *same* boosting + sparsification approach as in the outer loop, but this time with the weak hypotheses furnished by the non-robust PAC learner. As for result 2), the hard instance they exhibit is a simple binary classification problem over a universe of size 2 where only one of the elements x_0 in the universe has nontrivial perturbation set U(x_0) of size s, and the non-robust learner tries to learn a univariate threshold function consistent with a random embedding of the data on the real line. The non-robust learner is designed essentially to only give O(1) bits of information at every step, so one requires log(s) rounds of interaction to get zero robust error. Finally, as the authors acknowledge, result 3) is proven essentially by invoking a standard argument for passing between the mistake-bound model and the PAC model. POST-AUTHOR FEEDBACK: After some clarification about the line in the rebuttal about how a cheating reduction algorithm won't generalize in general, it seems that the authors have a technically stronger and conceptually more compelling lower bound that will appear in the full version. Because of this, as well as their commitment to making the paper more self-contained and cleaning up the writing, I feel that the authors have satisfactorily addressed all my primary concerns. While the result is purely of theoretical interest for the time being, it's a solid first step towards understanding robustifying wrappers. I'm upgrading to a 7 and vote for acceptance.

Strengths: The conceptual question being asked in this work is an extremely important one for the community to investigate. Also, while previous works have considered various versions of the question of robustifying non-robust learning algorithms in a black box fashion, the version this paper asks is more general and the bounds it gets are quantitatively superior. In particular, they make an important step towards resolving the dependence of the query complexity on the size of the perturbation sets, and the argument for the upper bound is a fairly simple but slick extension of the ideas from [MHS19].

Weaknesses: While this work takes an important step towards answering the question they set out to solve (in the robustly realizable case), the fact that you provably need the perturbation set to be finite (without assuming existence of an online learner) seems indicative of the fact that maybe this work isn't asking the right version of this question or that a black-box robustification procedure in practically relevant situations is too much to hope for. Concretely, one of my main complaints about the model is that the hard instance for the lower bound isn't particularly compelling, though I'm happy to be convinced otherwise. One way to get a robust PAC learner for their hard instance is just to ignore what the non-robust learners say and output 1 iff the input is x_1. This doesn't fall under their definition of an aggregation reduction, and granted, this is a slippery slope because if you start allowing the aggregator to do arbitrary computations on the side, we can't rule out that there just exist robust PAC learners for arbitrary robustly realizable problems that totally ignore what the non-robust learners tell them. It's not clear how to formulate the question in a meaningful way that avoids these issues, but it does feel like the main takeaway from the lower bound instance in this paper could also just be that one shouldn't try black-box robustifying an algorithm like Algorithm 2 to begin with because it's a bad algorithm for the learning problem at hand. In that sense, the question they raise in the conclusion of robustifying learning algorithms that are already *weakly robust* seems like a potentially far more interesting problem. Another complaint is that the gap between log^2 and log is quite large. For all L_inf bounded perturbations of an image, this is the difference between a linear or quadratic dependence in the dimension for the oracle complexity.

Correctness: I have convinced myself that the proofs are correct.

Clarity: While the proof of Theorem 3.1 relies heavily on [MHS19], I really think the authors should spend more time in the supplementary material fleshing out the details of the argument, even if it means writing some formal statements for some of the ingredients being used, like the Moran-Yehudayoff trick in lines 184-188, why it suffices to upper bound the VC and dual VC dimensions of the concept class that the predictors output by ZeroRobustLoss live in, maybe even the guarantees of alpha-boost. Admittedly these are all touched upon in the proof of Theorem 4 in [MHS19] to some extent, but given that the techniques in that paper are clearly broadly useful, giving a precise description of them in the supplementary material would put the overall proof on less shaky foundations. For the lower bound, maybe I'm misunderstanding, but it seems like there are some minor but confusing typos in the pseudocode, e.g. in the definition of M_1,M_2. For instance, one very dumb question I had was: why is M_1 as defined not just always equal to 1? If z is not in P^t and y = 0, isn't z by definition an element of {z_1,...,z_k} for which h_{t-1} outputs 0? My understanding of the proof is that you pick a random ordering for z_1,...,z_k at the beginning, and at every step the PAC learner either outputs the previous hypothesis because it already gets eps non-robust error, or you shift the threshold forward by as little as possible so that among the previously misclassified points, you correctly class 1 - eps fraction. Regardless of how the D_t's are set, an averaging argument says there will exist one choice of permutation that ensures you'll only make a bounded amount of progress at every step. Also, if this interpretation is correct, again, I don't see any harm in writing something informal along these lines prior to giving the formal proof, for the sake of readability.

Relation to Prior Work: Yes, the relations to relevant prior work are spelled out very clearly.

Reproducibility: Yes

Additional Feedback: Minor typos: - P. 1: "That is, we are asking whether *it's* possible" - P. 3: "if and only if *its* VC dimension is finite" - P. 4: "when *studying" reductions" - P. 6: "lower bonds" -> "lower bounds"


Review 4

Summary and Contributions: The paper studies the problem of using non-robust PAC learners to construct a PAC learner that is robust against adversarial perturbations. The authors establish several interesting results, including constructive arguments and lower bounds. First, they prove that when the learner is allowed access to the adversary U (which is a function from the domain X to 2^X that specifies the allowable perturbations of x), then there exists a robust learning algorithm that only relies on non-robust learners. In addition, the sample complexity of that algorithm is independent of |U| but its running time depends polynomially on log |U|. They also show that the dependence on log |U| is unavoidable without making further assumptions. Second, if the learner has only access to a mistake oracle, then the minimum number of queries to the non-robust learner can be as high as |U| without making any further assumptions. However, if we assume that the non-robust learner is an online learning algorithm with a finite mistake bound, which I think is a strong assumption, then the number of calls to the non-robust learner can be independent of |U|. The authors also analyze the sample complexity and running time of the algorithms. ======= Post Rebuttal: I am generally satisfied with the authors' response. One concern I had is about the realizable setting but we know in DNNs, in particular, that this is not a strong assumption. It is good to know that the authors believe they can extend their results to certain types of noise. After reading the other reviews and the authors' response, I have decided to keep my score.

Strengths: The paper presents several strong theoretical results, including constructive arguments, impossibility results, and matching lower bounds. It is well-written and a great care has been made on ensuring that the mathematical formalism is precise and easy to follow. Second, the topic is very important to the community. Understanding the level of robustness that can be achieved against adversarial perturbation is important given the growing applications of machine learning in very critical domains. In addition, the authors connect their results to the recent literature and raise some interesting open problems for future research.

Weaknesses: The emphasis on the paper has been on the size of the adversary |U|. The reason behind this emphasis on |U| is because |U| is often infinite in practice (e.g. balls in Lp spaces) or, still, extremely large if we take finite precision into account. Hence, one may deduce impossibility results for robust learning by establishing non-trivial dependence on |U|. So, it is important. On the other hand, for the results to be useful in practice, the upper bounds contain other important terms, such as the dual VC dimension, which can be exponentially larger than the VC dimension. This is important because the authors argue that the number of calls in Algorithm 1 is O(log m+ log |U|) but that notation suppresses the dependence on the dual VC dimension d*. The number of calls in Algorithm 1 grows at least linearly with d*, which can be exponential on the number of model parameters. So, the guarantee may not be useful in practice except for simple hypothesis spaces with a small dual VC dimension, such as linear classifiers. In addition, the present paper only studies the realizable setting. In practice, this is rarely true. It seems to me that the current proof technique using alpha-boost is not easy to extend to the agnostic setting but I might be wrong.

Correctness: The arguments appear to be correct to me. However, I could not understand the proof of Theorem 3.4 so I cannot judge if it is correct or not.

Clarity: The paper is well-written. The authors have taken a great care at formalizing the setting and the notation is easy to follow.

Relation to Prior Work: Yes, the relation between the new results and previous works is clearly discussed throughout the paper.

Reproducibility: Yes

Additional Feedback: In Line 233, the authors state that "We can then argue that if we cannot minimize the empirical robust loss by calling non-robust learner A, there is no hope for robust learning." I don't see how this statement fits with the rest of the paper. The reason I am mentioning this is that the authors show that impossibility results can sometimes be circumvented by imposing some additional assumptions on the non-robust learner, such as that it is an online learner. So, establishing that one cannot minimize the empirical robust loss by calling an *arbitrary* non-robust learner does not mean there is no hope for robust learning. it might only mean one has to choose the right non-robust learner.

[Author Response · NeurIPS 2020]

We thank the reviewers for their detailed comments and thoughtful feedback.

Following your feedback, we will work on making the paper easier to read and especially making it more self-contained:
we will include a full and self-contained proof of Theorem 3.1 (instead of relying on [MHS19]), add a description of
$\alpha$-Boost and lemmas stating its properties, and a lemma for the sparsification technique due to Moran-Yehudayoff.

Below we address other key concerns that were raised:

**Reviewer 1**: This paper studies what guarantees can we (theoretically) hope for in adversarially robust learning when
all we have access to is a black-box non-robust learner. This question helps us formulate two settings (Section 3 and
Section 4). Formally defining these settings can be helpful for practice, in the sense that they progress us towards the
non-trivial task of formulating the "right" access we need to allow/assume/use on perturbation sets. Such progress can
help us handle robust learning beyond $\ell_p$ perturbations, and obtain generic wrappers for non-robust learning for generic
perturbation sets. For instance, the mistake oracle model presented in Section 4 captures what's referred to in practice
as adversarial training, and can be applied for generic perturbation sets beyond $\ell_p$ perturbations.

In the statement of Theorem 4.3 (lines 317-320), we do explicitly specify the sample/oracle complexity of the resulting
robust learner, in terms of the mistake bound of the black-box learner we are reducing to (as in any reduction, the
guarantee on the wrapper is in terms of the guarantee of the method being wrapped–we hope this is clear). But we
agree it will be useful to include a concrete example of how this can be instantiated, and will do so.

**Reviewer 2**: Thank you for pointing us to the related work papers, we will add these citations to the paper. We briefly
discuss the agnostic setting in lines 335-342, and mention that the techniques in [MHS19] give us a reduction with an
exponential runtime dependence on VC dimension. Our results can also be extended to the Massart noise model. This
can be done with roughly similar sample/oracle complexity as in the realizable setting.

*Sampling Oracle Over Perturbations:* We need an oracle that takes as input a point $x$ and a function $E : \mathcal{X} \to \mathbb{R}$, and
does the following:

1. Samples a perturbation $z$ from a distribution given by $p_x(z) \propto \exp(E(z)) * \mathbb{1}[z \in \mathcal{U}(x)]$. That is, the oracle
samples from the set $\mathcal{U}(x)$ based on the weighting encoded in $E$.
2. Calculates $\Pr[z \in \mathcal{U}(x)]$ for the distribution given by $p(z) \propto \exp(E(z))$.

Using such an oracle suffices to sample from distributions over the inflated set $S_\mathcal{U}$ that are constructed by $\alpha$-Boost in
Algorithm 1 and its subprocedure `ZeroRobustLoss`. We will update the sampling paragraph (lines 287-295) in the paper
with a more elaborate/formal explanation of these details, and we will also address your other comments/corrections.

**Reviewer 3**: First, we think that reductions with a $\log |\mathcal{U}|$ dependence could, sometimes, be reasonable, as pointed out
by Reviewer 2. Second, since we agree that in many cases we want to avoid the $\log |\mathcal{U}|$ dependence, our work does
indeed indicate that this means moving away from reductions based on an explicit $\mathcal{U}$ oracle, which is exactly what we
do in Section 4 (using the mistake oracle model). We view this as a significant contribution of our work, since this isn't
obvious (at least to us) a-priori, or from [MHS19], and it progresses us towards the non-trivial task of formulating the
"right" access we need to allow/assume/use. Although the lower bound holds for aggregation reductions, a cheating
reduction algorithm that ignores the non-robust learners can't generalize in general, and we will make this formal
in the paper. Regarding the proof of the lower bound, your intuition and understanding is correct. We will add an
informal description of the strategy before the formal proof as suggested. In the definition of $M_1$, there is a missing
multiplicative factor of $\Pr_{(z,y)\sim D_t}[z \notin P^t \wedge y = 0]$, and $M_2$ should be $(1 - \varepsilon) - M_1$. Thank you for pointing out
these typos. Basically, $M_1$ is the fraction of examples correctly classified under distribution $D_t$ by predictor $h_{t-1}$. If
$M_1$ is high enough (at least $1 - \varepsilon$), then it suffices to return $h_{t-1}$, if not, we shift the threshold as little as possible based
on $M_2$.

**Reviewer 4**: While true that the oracle complexity can be exponential in the VC dimension, in lines 107–109 we
discuss how for many natural classes, the dual VC dimension is polynomially related to the VC dimension. This holds
beyond just linear predictors, and is true also, e.g., for neural networks with threshold activation functions. This can
be shown using Lemma 3.2 which is a novel contribution in this paper. We briefly discuss limitations/challenges of
extending our results to the agnostic setting in lines 335-342. Our results can also be extended to the Massart noise
model. This can be done with roughly similar sample/oracle complexity as in the realizable setting. Regarding line 233,
yes, it does NOT mean that there is no hope for robust learning in general. We are just showing that $\log |\mathcal{U}|$ oracle calls
are unavoidable without extra assumptions on the non-robust learner.

[Meta-Review · NeurIPS 2020]

All the reviewers agreed that the paper provides a novel and important theoretical result. Specifically, one of the main contributions of the paper is to show that if a problem is PAC learnable, i.e., has finite VC dimension d, then it i also robustly PAC learnable. The result had already been proven last year, however, this paper provides a novel framework to prove it by showing how to reduce the task of robust learning to the task of normal PAC learning (it should be noted that the reduction may not be efficient (i.e., polynomial time) but still is important from the theoretical and statistical perspective. The reviewers also had some suggestions for the revised version of the paper which are reflected in the'r updated reviews.